# Bacterial interspecies interactions modulate pH-mediated antibiotic tolerance

Andrés Aranda-Díaz[1], Benjamin Obadia[2], Ren Dodge[3], Tani Thomsen[1], Zachary F Hallberg[4], Zehra Tüzün Güvener[2], William B Ludington[2,3]*, Kerwyn Casey Huang[1,5,6]*

[1]Department of Bioengineering, Stanford University, Stanford, United States; [2]Department of Molecular and Cell Biology, University of California, Berkeley, Berkeley, United States; [3]Department of Embryology, Carnegie Institution of Washington, Baltimore, United States; [4]Department of Plant and Microbial Biology, University of California, Berkeley, Berkeley, United States; [5]Department of Microbiology and Immunology, Stanford University School of Medicine, Stanford, United States; [6]Chan Zuckerberg Biohub, San Francisco, United States

**Abstract** Predicting antibiotic efficacy within microbial communities remains highly challenging. Interspecies interactions can impact antibiotic activity through many mechanisms, including alterations to bacterial physiology. Here, we studied synthetic communities constructed from the core members of the fruit fly gut microbiota. Co-culturing of *Lactobacillus plantarum* with *Acetobacter* species altered its tolerance to the transcriptional inhibitor rifampin. By measuring key metabolites and environmental pH, we determined that *Acetobacter* species counter the acidification driven by *L. plantarum* production of lactate. Shifts in pH were sufficient to modulate *L. plantarum* tolerance to rifampin and the translational inhibitor erythromycin. A reduction in lag time exiting stationary phase was linked to *L. plantarum* tolerance to rifampicin, opposite to a previously identified mode of tolerance to ampicillin in *E. coli*. This mechanistic understanding of the coupling among interspecies interactions, environmental pH, and antibiotic tolerance enables future predictions of growth and the effects of antibiotics in more complex communities.

*For correspondence:
ludington@carnegiescience.edu (WBL);
kchuang@stanford.edu (KCH)

**Competing interests:** The authors declare that no competing interests exist.

## Introduction

Decades of investigations have described detailed and precise molecular mechanisms of antibiotic action using model organisms such as *Escherichia coli* grown in monoculture. However, unlike in laboratory monocultures, the vast majority of bacteria live in diverse communities. In environments such as the human intestine, antibiotics impact communities in many ways, ranging from the loss of diversity (*Dethlefsen et al., 2008*; *Jernberg et al., 2007*) to the evolution of multidrug-resistant pathogens (*Santajit and Indrawattana, 2016*). Hence, there is a pressing need for new frameworks that predict how antibiotics affect bacterial communities in ways that cannot be predicted from simple monocultures.

Synthetic microbial communities provide the opportunity to perform controlled experiments that uncover mechanistic principles of microbial ecology (*Momeni et al., 2011*; *Zuk and Travisano, 2018*). Such communities have been used to demonstrate the evolution and consequences of microbial cooperation (*Harcombe, 2010*; *Momeni et al., 2013*), reveal the role of interactions on spatial patterning (*Momeni et al., 2013*), and to link physiology and metabolism to mutualistic relationships (*Ponomarova et al., 2017*). Interspecies interactions can occur through specific mechanisms involving members of a community (e.g. cross-feeding or competition for resources) and through global

environmental variables modified by bacterial activity. An example of the latter is pH, which has recently been shown to drive community dynamics in an artificial laboratory system of decomposition bacteria (*Ratzke and Gore, 2018*).

A natural system with low diversity provides a consortium of species with a common evolutionary and ecological history from which to build synthetic communities (*Ponomarova et al., 2017*). The gut microbiota of *Drosophila melanogaster* fruit flies is such a low diversity assemblage in which all members can be cultured in vitro (*Obadia et al., 2017*), making it amenable to the systematic dissection of bacterial interactions. This community consists of ~5 species predominantly from the *Lactobacillus* and *Acetobacter* genera (*Wong et al., 2011*) (*Figure 1A*). The metabolic lifestyles of species from these genera in isolation have been largely uncovered. *Lactobacillus* spp. produce lactic acid (*Makarova et al., 2006*), while *Acetobacter* spp. produce acetic acid and are distinguished by their ability to oxidize lactate to carbon dioxide and water (*Yamada et al., 1997*).

Bacteria can survive antibiotics through (i) resistance mutations, which counteract the antibiotic mechanism and increase the minimum inhibitory concentration (MIC); (ii) tolerance, whereby the entire population enters an altered physiological state that prolongs survivability without changing the MIC of the antibiotic, leading to an increase in the time required to kill a given fraction of the population; (iii) heteroresistance, whereby a subset of the population has a higher MIC and grows at concentrations that would otherwise kill the population; and (iv) persistence, whereby a subset of the population survives treatment for a longer period (*Balaban et al., 2019*; *Brauner et al., 2016*). Members of multispecies communities, such as biofilms and models of urinary tract infections, can display altered sensitivity to antibiotics in the community context (*Adamowicz et al., 2018*; *de Vos et al., 2017*; *Nicoloff and Andersson, 2016*; *Sanchez-Vizuete et al., 2015*). Previous studies have used synthetic communities to uncover the interplay between interspecies interactions and antibiotic efficacy; for example, the exoproducts of *Pseudomonas aeruginosa* affect the survival of *Staphylococcus aureus* through changes in antibiotic uptake, cell-wall integrity, and intracellular ATP pools (*Radlinski et al., 2017*). In genetically modified communities, intracellular antibiotic degradation affords cross-species protection against chloramphenicol (*Sorg et al., 2016*). Additionally, metabolic dependencies within synthetic communities can lower the viability of bacteria when antibiotics eliminate providers of essential metabolites, leading to an apparent change in the MICof the dependent species (*Adamowicz et al., 2018*). However, we still lack understanding of how metabolic interactions between bacteria affect the physiological processes targeted by antibiotics and the resulting balance between growth inhibition (bacteriostatic activity) and death (bactericidal activity). For example, the intimate relationship between bacterial metabolism and environmental pH could also lead to changes in antibiotic efficacy, as previously shown in monocultures (*Aagaard et al., 1991*; *Argemi et al., 2013*; *Kamberi et al., 1999*; *Karslake et al., 2016*; *Yang et al., 2014*). The interplay of all these processes in complex communities will provide new ways to combat pathogen survival and resistance evolution, particularly in cases involving tolerance, an important and understudied aspect of antibiotic susceptibility that can be elicited by diverse mechanisms and can facilitate the evolution of resistance (*Levin-Reisman et al., 2017*).

In this study, we interrogated how interspecies interactions affect growth, pH, and antibiotic susceptibilities. We used high-throughput assays to measure and compare these parameters in monocultures and co-cultures. We found that *Lactobacillus plantarum* (*Lp*) exhibited antibiotic tolerance (delay in death *Brauner et al., 2016*) in the presence of *Acetobacter* species. Lactate accumulation by *Lp* in monocultures acidified the media, inhibiting growth during stationary phase. *Acetobacter*-mediated lactate consumption released this inhibition by increasing pH, leading to a shorter *Lp* lag while exiting stationary phase. This reduced lag exiting stationary phase was correlated with the antibiotic tolerance of *Lp* that we observed. We determined that changes in pH elicited by *Acetobacter* activity were sufficient to modulate tolerance of *Lp* to both rifampin and erythromycin. Taken together, our findings indicate that simple changes to the environment can drive complex physiological behaviors and antibiotic responses within bacterial communities.

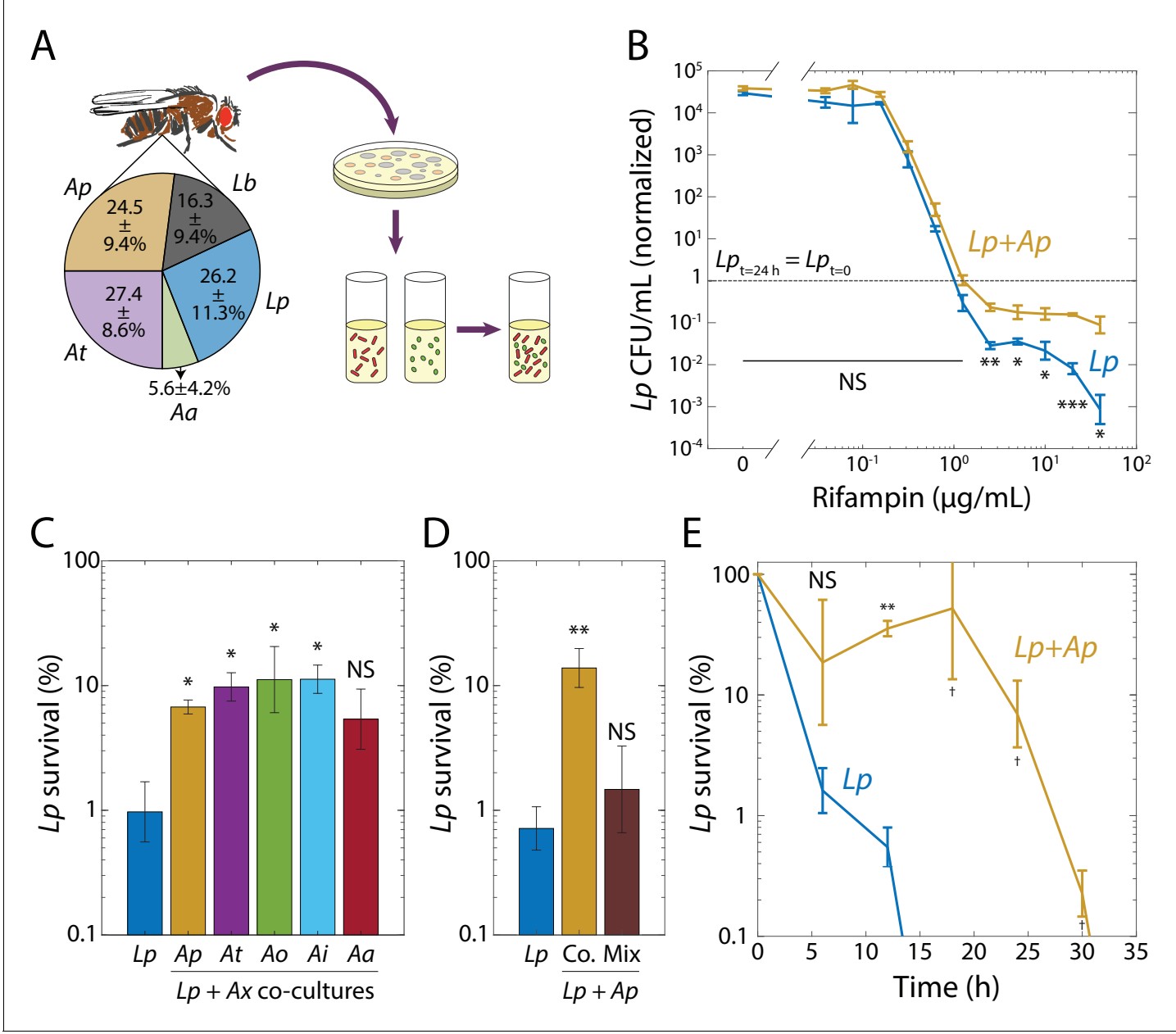

**Figure 1.** Interspecies interactions in synthetic communities derived from the fruit fly gut microbiome induce rifampin tolerance. **(A)** Synthetic community construction scheme. Relative abundances of the dominant species in the *D. melanogaster* gut microbiome determined from 16S rRNA sequencing. All reads mapped to two *Lactobacillus* species: *L. plantarum* (*Lp*) and *L. brevis* (*Lb*), and to three *Acetobacter* species: *A. pasteurianus* (*Ap*), *A. tropicalis* (*At*), and *A. aceti* (*Aa*). Values are mean ± standard deviation (S.D.), $n = 18$. Mean and S.D. were weighed by the total number of reads for each fly. Strains were isolated in agar plates and grown in liquid culture. Synthetic communities were then built by co-culturing individual species in liquid culture. **(B)** When grown with *Ap*, *Lp* survived after 24 h at rifampin concentrations above the MIC. Viable cell plating counts of *Lp* after growth in rifampin for 24 h normalized to the counts at the start of the experiment ($t = 0$, $3 \times 10^5$ and $4 \times 10^5$ CFU/mL in monoculture and co-culture, respectively). Dashed line indicates where the population at $t = 24$ h would have the same size as at $t = 0$. Error bars are S.D. for each condition, $n = 3$. p-values are from a Student's two-sided *t*-test of the difference of the co-culture from the monoculture (*: $p<4\times10^{-3}$, **: $p<8\times10^{-4}$, ***: $p<8\times10^{-5}$, equivalent to *: $p<0.05/n$, **: $p<0.01/n$, ***: $p<1\times10^{-3}/n$, where $n = 12$ is the number of comparisons; NS: not significant). **(C)** Protection of *Lp* at supra-MIC concentrations of rifampin is elicited by all *Acetobacter* species tested. CFU/mL of *Lp* grown in monoculture (*Lp*) or in co-culture with *Ap*, *At*, *A. orientalis* (*Ao*), *A. indonesiensis* (*Ai*), and *Aa*, and then treated with 20 μg/mL rifampin for 24 h normalized to counts at the start of the experiment ($t = 0$). Error bars are S.D. for each condition, $n = 3$. p-values are from a Student's two-sided *t*-test of the difference from the monoculture (*: $p<0.01$, equivalent to *: $p<0.05/n$, where $n = 5$ is the number of comparisons; NS: not significant). **(D)** *Ap*-mediated survival of *Lp* at rifampin concentrations above the MIC is history-dependent, requiring co-culturing before exposure as compared with mixing. CFU/mL of *Lp* grown in monoculture, in co-culture with *Ap* (Co.), or mixed with *Ap* without subsequent growth in the absence of antibiotic (Mix), and treated with 20 μg/mL rifampin for 24 h

*Figure 1 continued on next page*

*Figure 1 continued*

normalized to counts at the start of the experiment ($t = 0$). Error bars are S.D. for each condition, $n = 3$. p-vValues are from a Student's two-sided *t*-test of the difference from the monoculture (**: $p<5\times10^{-3}$, equivalent to **: $p<0.01/n$, where $n = 2$ is the number of comparisons; NS: not significant). (**E**) The time to killing of *Lp* under rifampin treatment is extended in the presence of an *Acetobacter*. CFU/mL of *Lp* grown in monoculture and co-cultured with *Ap*, and treated with 50 µg/mL rifampin, normalized to counts at the start of the experiment ($t = 0$). Error bars are S.D. for each condition, $n = 3$. p-values are from a Student's two-sided *t*-test of the difference from the monoculture at the corresponding timepoint (**: $p<2\times10^{-3}$, equivalent to **: $p<0.01/n$, where $n = 5$ is the number of comparisons; †: measurement below limit of detection; NS: not significant). Values off the graph were below the limit of detection of the assay.

The online version of this article includes the following figure supplement(s) for figure 1:

**Figure supplement 1.** Growth curves of monocultures of the primary species in the fruit fly gut microbiota measured with a plate reader (Materials and methods).

**Figure supplement 2.** Changes in the survivability of *Lactobacillus plantarum* (*Lp*) when co-cultured with *Acetobacter pasteurianus* (*Ap*) are not due to increased survivability of *Ap*, inoculum size, or *Lp* growth in rifampin.

## Results

### Interspecies interactions induce tolerance to rifampin in a synthetic community

To determine the composition of the gut microbiota in our laboratory fruit flies, we performed 16S rRNA sequencing from 18 individual dissected guts (Materials and methods). We identified five species belonging to seven unique operational taxonomic units (OTUs) by clustering the sequences at 99% identity: *L. plantarum* (*Lp*), *L. brevis* (*Lb*), *Acetobacter pasteurianus* (*Ap*), *A. tropicalis* (*At*), and *A. aceti* (*Aa*) (*Figure 1a*). We then isolated the species in culture (*Supplementary file 1a*) and determined the antibiotic sensitivities of the four major fly gut inhabitants (*Lp*, *Lb*, *Ap*, and *At*; *Figure 1A*) in vitro grown in Man, Rogosa, and Sharpe (MRS) medium.

We tested 10 antibiotics representing a wide variety of classes using plate-based growth assays (Materials and methods). For many drugs, some of the isolates were resistant (detectable growth) at least up to the highest concentrations tested. Rifampin was the only drug for which all four species exhibited sensitivity (*Supplementary file 1b*) and it is bactericidal (*Walsh and Wencewicz, 2016*), hence there is the opportunity to study survival as well as sensitivity.

Since *Lb* grew significantly more slowly and had a much longer lag phase than *Lp* (*Figure 1—figure supplement 1*), we focused on *Lp* and its interactions with the *Acetobacter* species, particularly *Ap*, which is more abundant in the fly gut than the other *Acetobacter* species (*Figure 1A*). We grew *Lp* and *Ap* separately for 48 h in test tubes, combined them in test tubes at an optical density at 600 nm (henceforth OD) of 0.02 each, and co-cultured them in MRS for 48 hr. We then diluted this co-culture and 48 h monocultures of *Lp* and *Ap* into fresh MRS at ~$5 \times 10^5$ colony-forming units/mL (CFU/mL) in 96-well plates over a range of rifampin concentrations. To determine whether the co-culture still contained both species, we measured the percentage of survival and the fraction of each species at various rifampin concentrations by plating on selective media and exploiting species-specific colony morphologies (*Supplementary file 1a*). In the co-culture, *Ap* died off at a similar concentration of rifampin as during growth in a monoculture (1.25 µg/mL, *Figure 1—figure supplement 2A*). For *Lp*, the MIC was similar in co-culture as in monoculture (1.25 µg/mL, *Figure 1B*), but at concentrations above the MIC, significantly more *Lp* cells survived in co-culture versus monoculture (*Figure 1B*). This effect could not be explained by small differences in the initial inoculum, as increasing cell densities up to 100-fold did not change the MIC or survival of *Lp* in monoculture (*Figure 1—figure supplement 2B,C*). Because of the change in *Lp* survival, we focused herein on this phenotype.

To determine whether *Lp*'s increased survival was specific to co-culturing with *Ap*, we co-cultured *Lp* with each of the *Acetobacter* species, including a wild fly isolate of *A. indonesiensis* (*Ai*), and lab fly isolates of *A. orientalis* (*Ao*) and *Aa*, the fifth member of the microbiota of our flies (*Figure 1A*). We then diluted each co-culture to an initial *Lp* cell density of ~$5 \times 10^5$ CFU/mL into fresh MRS with 20 µg/mL rifampin (16X MIC) and let the cells grow for 24 hr. Co-culturing with any of the *Acetobacter* species increased survival by approximately one order of magnitude (*Figure 1C*). To determine whether this increased survival requires co-culturing prior to rifampin treatment (rather than the presence of *Acetobacter* species being sufficient), we grew *Lp* and *Ap* separately for 48 h and

mixed and diluted them at the time of addition of 20 µg/mL rifampin. After 24 h of incubation, the number of CFU/mL was significantly lower in mixed culture than in co-culture (*Figure 1D*), indicating a history dependence to increased survival.

To determine whether co-culturing slows killing by the drug, we examined the survival of *Lp* over time at a high drug concentration (50 µg/mL, 40X MIC). Similar to the experiments above, we compared *Lp* CFU/mL in a monoculture with that in a co-culture with *Ap*. In monoculture, *Lp* rapidly died, with CFU/mL becoming undetectable within 18 hr; by contrast, *Lp* survived >30 h after co-culturing (*Figure 1E*). To test whether the increased time to death was due to a disruption in the balance of growth and death, we performed single-cell imaging of mono- and co-cultures on MRS agarose with 50 µg/mL rifampin. We observed no growth in either mono- or co-cultures over 3 h of imaging ($n = 568$ and $n = 236$ cells, respectively; *Figure 1—figure supplement 2D*), indicating that survivability is due to protection from death, as opposed to increased growth. Because the MIC of *Lp* was unchanged in co-cultures compared with monocultures (no resistance), and the delay in killing was observed in the bulk population in co-cultures, these results indicate that co-culturing *Lp* with *Ap* induces tolerance of *Lp* to rifampin (*Balaban et al., 2019*; *Brauner et al., 2016*).

## Co-culturing leads to growth of *Lp* in stationary phase

To investigate possible environmental factors that could be linked to *Lp* tolerance to rifampin when co-cultured with *Acetobacter* species (*Figure 1C,E*), we first inquired whether the total amount of growth of the co-culture was larger or smaller than expected from the yield of the monocultures. We grew *Lp* and each of the *Acetobacter* species separately for 48 hr, diluted the monocultures to OD = 0.04, combined the *Lp* monoculture 1:1 with each *Acetobacter* monoculture, and grew the co-cultures for 48 h in a plate reader. We then diluted the saturated cultures 1:30 in phosphate-buffered saline (PBS) to accurately measure the final OD, and computed an interaction score based on an additive model:

$$\alpha = \frac{\mathrm{OD_{co}} - \left(\mathrm{OD}_{Lp} + \mathrm{OD}_A\right)}{\sqrt{\mathrm{OD}_{Lp}\mathrm{OD}_A}} \qquad (1)$$

where $\mathrm{OD_{co}}$ and $\mathrm{OD}_A$ are the final ODs of the co-culture and the *Acetobacter* monoculture, respectively. With this metric, $\alpha > 0$ indicates synergy and $\alpha < 0$ indicates antagonism. *Ap* showed a strong (positive) interaction with *Lp*, whereas *At*, *Ai*, and *Aa* had an interaction score closer to 0 (*Figure 2—figure supplement 1A*). Only the *Lp-Ao* co-culture was significantly antagonistic (*Figure 2—figure supplement 1A*). We also performed this measurement in cultures grown in test tubes, where we only observed a significant positive interaction of *Lp* with *Ap* (*Figure 2—figure supplement 1A*). We then determined the total carrying capacity of each species in the co-cultures by counting CFUs. The *Lp* CFU/mL values for 48-h co-cultures with *Ap*, *At*, and *Ai* were higher than the *Lp* monoculture (*Figure 2A*); *Ap* showed the strongest effect. *Aa* and *Ao* did not significantly increase *Lp* CFU/mL (*Figure 2A*). *At* and *Ao* reached lower CFU/mL in co-cultures with *Lp* than in monocultures, while *Ap*, *At*, and *Aa* did not change significantly (*Figure 2—figure supplement 1C*). Thus, *Lp* benefits from growth with certain *Acetobacter* species including *Ap*, and it has negative or neutral effects on the *Acetobacter* species (*Figure 2—figure supplement 1A, B*).

To determine when the additional *Lp* growth took place, we monitored CFU/mL values for *Lp* and *Ap* in co-culture throughout a 48 h time course starting from an initial combined cell density of $\sim 5 \times 10^5$ CFU/mL. Initially, *Lp* accounted for the bulk of the growth in the co-culture (*Figure 2B*). Interestingly, *Ap* in liquid monoculture showed little to no growth in most replicates after 40 h (*Figure 2B*); by contrast, in co-culture *Ap* started to grow after ~20 h and reached saturation by ~40 h (*Figure 2B*), indicating that *Ap* also benefited from growth as a co-culture. Thus, a mutualism exists between *Lp* and *Ap* driven by growth during stationary phase.

## Lactate metabolism leads to changes in pH in stationary phase co-cultures

Interestingly, after 30 hr, *Lp* displayed a significant (~2X) increase in CFU/mL in the *Lp-Ap* co-culture that did not occur in the monoculture (*Figure 2B*), indicating that the increase in final yield occurs late in stationary phase. Since this increase occurs generally (*Figure 2B*), we hypothesized that *Lp* has a common metabolic interaction with each of the *Acetobacter* species. An obvious candidate is

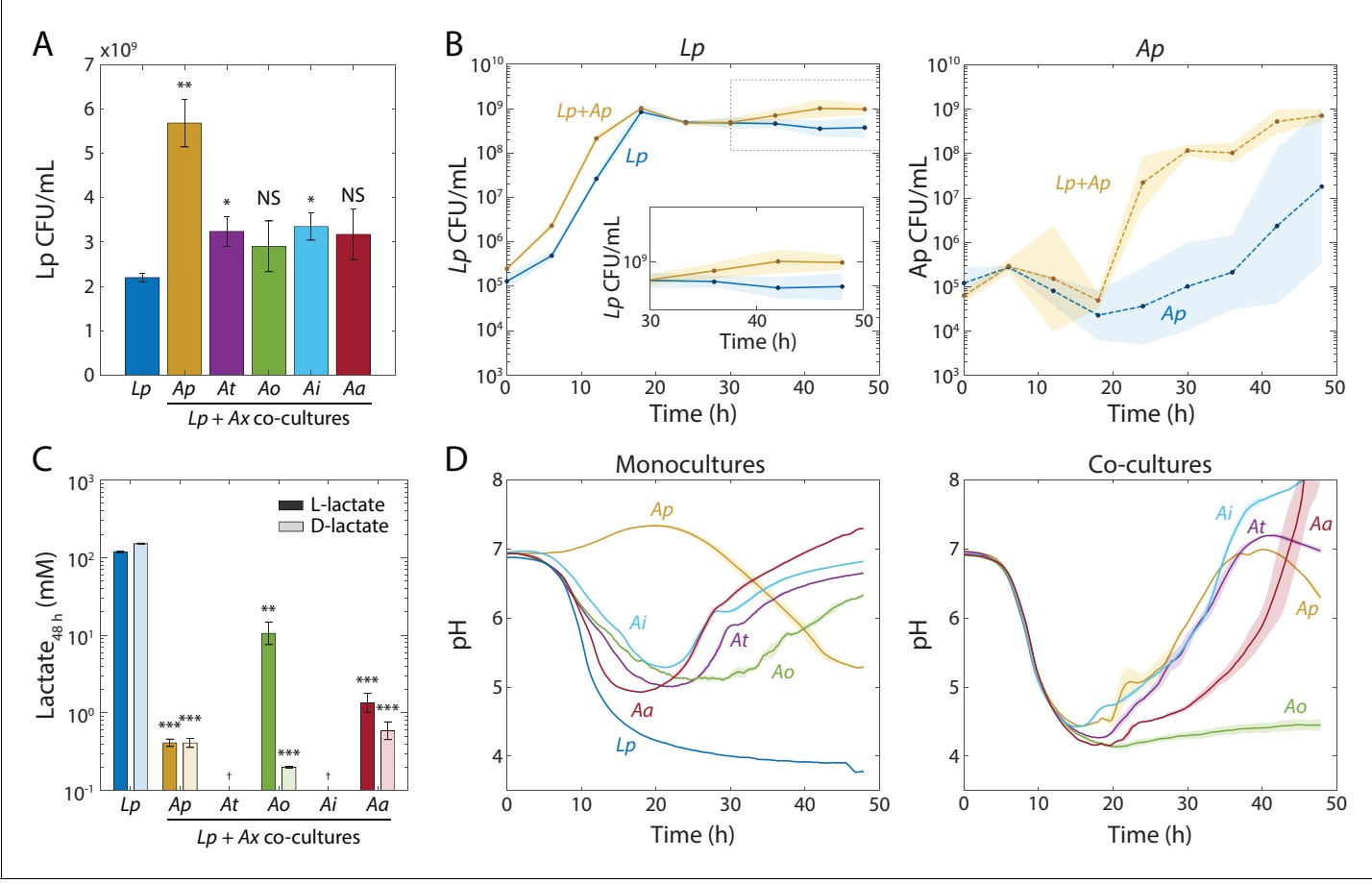

**Figure 2.** *Lp* growth during stationary phase in *Acetobacter* co-cultures is associated with an increase in pH and a decrease in lactate concentration. (**A**) Co-culturing *Lp* with *Ap*, *At*, or *Ai* resulted in increased *Lp* cell density after 48 hr. Co-culturing with *Ao* or *Aa* did not significantly increase *Lp* cell density by 48 hr. Error bars are standard deviation (S.D.) for each condition, $n = 3$. p-values are from a Student's two-sided *t*-test of the difference from the monoculture (*: $p<0.01$, **: $p<2\times10^{-3}$, equivalent to *: $p<0.05/n$, **: $p<0.01/n$, where $n = 5$ is the number of comparisons; NS: not significant). (**B**) Co-culturing *Lp* with *Ap* resulted in higher *Lp* cell density in stationary phase, as well as faster growth and shorter lag for *Ap*. Shaded regions indicate S. D., $n = 3$. Inset: zoom-in on region inside dashed box highlighting increase in carrying capacity in co-culture. (**C**) L- and D-lactate accumulated to much higher levels in *Lp* monocultures than in co-cultures after 48 h of growth, suggesting that *Acetobacter* spp. consumed *Lp*-produced lactate. Lactate concentration was measured enzymatically from culture supernatants at 48 hr. Error bars are S.D. for each condition, $n = 3$. p-values are from a Student's two-sided *t*-test of the difference from the monoculture (**: $p<2\times10^{-3}$, ***: $p<2\times10^{-4}$, equivalent to **: $p<0.01/n$, ***: $p<1\times10^{-3}/n$, where $n = 5$ is the number of comparisons; †: measurement below limit of detection). (**D**) The increase in *Lp* cell density in stationary phase is associated with an *Acetobacter*-dependent increase in pH early in stationary phase. pH was measured with the pH-sensitive dye 2',7-bis-(2-carboxyethyl)—5-(and-6)-carboxyfluorescein over time (Materials and methods). Shaded regions indicate S.D., $n = 3$. Data points were taken every 8.3 min.

The online version of this article includes the following figure supplement(s) for figure 2:

**Figure supplement 1.** Interaction scores of *Lactobacillus plantarum* (*Lp*) and *Acetobacter* spp.

**Figure supplement 2.** Lactate accumulates in *L. plantarum* (*Lp*) monocultures and co-cultures, and *Acetobacter*-driven consumption of lactate reverses the initial pH decrease.

**Figure supplement 3.** *L. plantarum* (*Lp*) intracellular pH is higher in stationary phase when grown with *Acetobacter pasteurianus* (*Ap*).

cross-feeding, since *Lp* produces lactate and the *Acetobacter* species consume it. We measured lactate levels in the supernatants of *Lp* monocultures and co-cultures of *Lp* with each of the *Acetobacter* species individually, after 48 h of growth. As expected, the *Lp* monoculture accumulated L- and D-lactate to high levels (>100 mM; **Figure 2C**). All co-cultures had significantly lower concentrations of both isomers than the monoculture (<2 mM, **Figure 2C**). The *Lp-Ao* co-culture harbored higher levels of L-lactate than any other co-culture and *Lp-Aa* had the highest concentration of L-lactate of the co-cultures (**Figure 2C**). *Lp-Ap*, *Lp-Ao*, and *Lp-Aa* co-cultures all accumulated lactate to >10 mM by 20 h (**Figure 2—figure supplement 2A**). Taken together, these data suggest that *Lp*

metabolism leads to an initial accumulation of lactate and that the *Acetobacter* species consume it, although *Aa* and *Ao* are less efficient at consuming L-lactate than the other species.

Since lactate is a short-chain fatty acid with a p$K_a$ of 3.86, we suspected that lactate production would affect the pH of the culture. We first monitored the pH dynamics of monocultures of *Lp* and of each *Acetobacter* species using the pH-sensitive fluorophore 2',7-bis-(2-carboxyethyl)−5-(and-6)-carboxyfluorescein (BCECF) (*James-Kracke, 1992*). In the *Lp* monoculture, pH decreased from pH = 6.75 to below four during growth (*Figure 2D*); more precisely, we measured a final supernatant pH = 3.77 using a pH meter (*Figure 2—figure supplement 2A*). For monocultures of all *Acetobacter* species except *Ap*, the medium first acidified down to pH ~5, and then increased back to pH = 6–7 (*Figure 2D*).

To test whether *Acetobacter* species reverse the pH decrease due to the accumulation of lactate produced by *Lp*, we measured the pH of co-cultures over time using BCECF. Co-cultures with *Ap*, *At*, and *Ai* followed similar trajectories in which the pH mimicked that of the *Lp* monoculture for the first 20 hr, after which the pH increased up to a final value of ~7 (*Figure 2D*). The *Lp-Aa* co-culture experienced a ~ 10 h delay in the pH increase, while the co-culture with *Ao* showed only a slight pH increase by 48 h (*Figure 2D*). The slight pH increase in the *Lp-Ao* co-culture is consistent with lower L-lactate consumption by *Ao* (*Figure 2C*). Using a pH meter for validation, we measured final pH values of 5.9, 5.8, 4.6, 5.4, and 4.8 in co-cultures with *Ap*, *At*, *Ao*, *Ai*, and *Aa*, respectively (*Figure 2—figure supplement 2B*). Thus, lactate metabolism dictates dramatic shifts in environmental pH that are related to physiological changes in antibiotic tolerance (*Figure 1C*).

Given the strong acidification of the medium in *Lp* monoculture but not in *Lp-Ap* co-culture (*Figure 2D*), we hypothesized that intracellular pH decreases in monoculture and increases in co-culture. To measure intracellular pH, we transformed our *Lp* strain with a plasmid expressing pHluorin (a GFP variant that acts as a ratiometric pH sensor *Martinez et al., 2012*) under the control of a strong constitutive promoter (*Rud et al., 2006*). The two absorbance peaks, which we measured at 405 and 475 nm, are sensitive to pH and the ratio of the emission (at 509 nm) at these two excitation wavelengths can be used to estimate intracellular pH. We grew this strain in monoculture and in co-culture with *Ap* and measured fluorescence over time in a plate reader. Because of the high auto-fluorescence of the medium at 405 nm (data not shown), we could only track changes in fluorescence at an excitation wavelength of 475 nm. We observed an initial increase in signal as the *Lp* cells started to proliferate (*Figure 2—figure supplement 3A*). After the cultures saturated ($t \sim 20$ hr), we detected a decrease in the signal down to the levels of medium autofluorescence in the monoculture (*Figure 2—figure supplement 3A*). In the co-culture, in which the extracellular pH was raised by the metabolic activity of *Ap*, fluorescence did not decrease over time (*Figure 2—figure supplement 3A*), suggesting that intracellular pH decreases over time in monoculture but not in co-culture.

To verify that the decrease in fluorescence in monoculture was due to a drop in intracellular pH, as opposed to a decrease in protein concentration, we sampled cells after 48 h of growth, centrifuged them, resuspended them in PBS to measure pHluorin signal at both its excitation wavelengths, and measured fluorescence within 1 min of resuspension. The pHluorin signal ratio was significantly higher in *Lp-Ap* co-culture than in an *Lp* monoculture (*Figure 2—figure supplement 3B*). Taken together, these data indicate that the intracellular pH of *Lp* cells is significantly lower in monoculture than in co-culture with *Acetobacter* species.

## Low pH inhibits the growth of *Lp* and extends lag phase

The growth rates of lactic acid bacteria such as *Lp* are known to be affected by pH (*Stecka and Grzybowsii, 2000*). Since *Ap* growth causes a large increase in the extracellular pH of an *Lp-Ap* co-culture, we sought to determine the general dependence of *Lp* growth properties on pH. We diluted a 48 h culture of *Lp* cells grown in MRS at starting pH = 6.75 to a starting OD = 0.02 in MRS adjusted to starting pH ranging from 3 to 8 (Materials and methods). We then measured growth and BCECF fluorescence (*Figure 3—figure supplement 1A,B*). For lower starting pH values, the carrying capacity was lower (*Figure 3A*) and varied over a large OD range from <0.02 to>2. For all starting pH values, *Lp* cells reduced the pH to a common final value of ~3.7 (*Figure 3A*). The bulk growth rate approached zero (*Figure 3B*) as the pH approached its final value, explaining the differences in yield. Interestingly, the maximum growth rate was also pH-dependent (*Figure 3C*), with the highest growth rate at starting pH = 7.

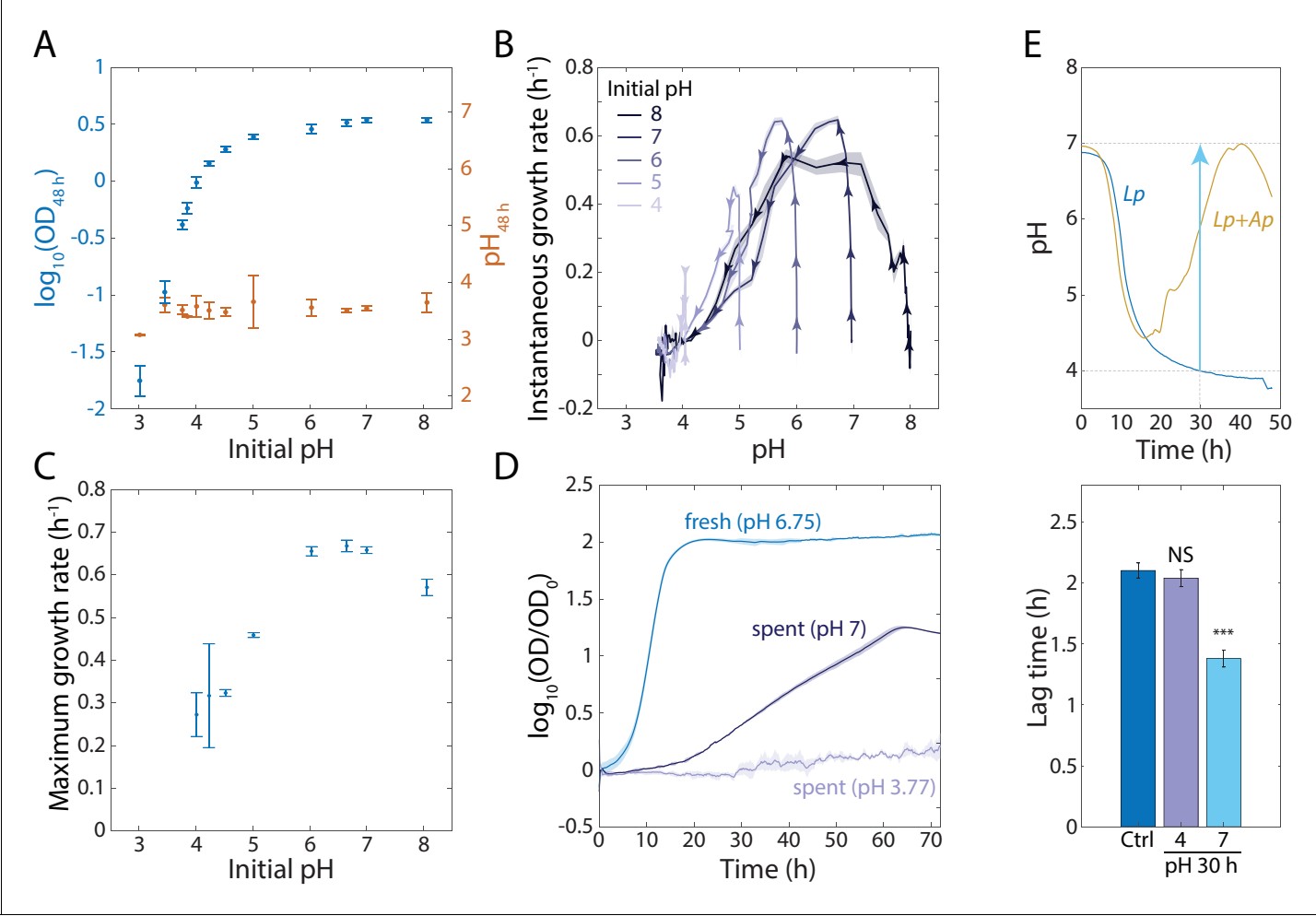

**Figure 3.** An increase in extracellular pH in stationary phase releases growth inhibition in *Lp* monocultures and shortens lag phase. **(A)** *Lp* growth is inhibited by low pH. Logarithm of OD (blue) and pH measured using BCECF (orange) after 48 h of growth in MRS at various starting pH values. Error bars are standard deviation (S.D.), *n* = 4. **(B)** Instantaneous growth rate in MRS is strongly linked to pH. Each curve was initialized at a different starting pH and represents 48 h of growth. Arrowheads indicate direction of time. Shaded regions are S.D., *n* = 4. Data points were taken every 8.3 min. **(C)** Maximal growth rate in MRS increases with increasing initial pH. Error bars are S.D., *n* = 4. **(D)** Increasing the pH of a saturated, spent *Lp* culture from 3.77 to 7 allows growth, although not as much as fresh MRS. Error bars are S.D., *n* = 3. Data points were taken every 8.3 min. **(E)** Increasing the pH of an *Lp* monoculture at *t* = 30 h from 4 to 7 to mimic the pH increase in *Lp-Ap* co-culture (top) leads to a shorter lag phase (bottom). Lag time was calculated by fitting growth curves to the Gompertz equation. Error bars are S.D., *n* = 3. p-values are from a Student's two-sided *t*-test of the difference from the control (***: $p < 5 \times 10^{-4}$, equivalent to ***: $p < 1 \times 10^{-3}/n$, where *n* = 2 is the number of comparisons; NS: not significant).

The online version of this article includes the following figure supplement(s) for figure 3:

**Figure supplement 1.** Growth of *L. plantarum* (*Lp*) in stationary phase is limited by the acidity of the medium.

---

Given these findings, we hypothesized that the inhibition of growth in stationary phase of an *Lp* monoculture is due to the decreased intracellular and extracellular pH, and that *Ap* releases *Lp*'s growth inhibition by raising intracellular and extracellular pH. To test this hypothesis, we inoculated a 48 h culture of *Lp* to an initial OD = 0.02 into the supernatant of a 48 h *Lp* culture at pH = 3.77 or manually adjusted to pH = 7. Almost no growth took place in supernatant starting from pH = 3.77, but we observed substantial growth (~20 fold increase in OD) of the bulk culture in supernatant raised to pH = 7 (**Figure 3D**). As expected, the maximal growth rate was lower than in fresh MRS, presumably due to the partial depletion of nutrients (**Figure 3D**); addition of glucose to the conditioned medium supported faster growth, but only starting from neutral pH (**Figure 3—figure supplement 1C**). We also hypothesized that the accumulation of lactate by *Lp* would allow growth of *Ap* in *Lp*-conditioned medium even at low starting pH. When we diluted a saturated *Ap* culture in

*Lp*-conditioned medium generated as above, *Ap* achieved an even higher OD than in fresh MRS (*Figure 3—figure supplement 1D*).

These findings suggested that *Ap*'s effects on *Lp* growth might be due primarily to the pH changes that *Ap* initiates because *Acetobacter* species can grow at low pH and consume lactate. Thus, we first increased the pH of an *Lp* monoculture to 7 after 30 h of growth, which is when the pH increased most rapidly in the *Lp-Ap* co-culture (*Figure 3E*). We then incubated the cells for an additional 18 hr. We did not observe a significant increase in CFUs/mL from this pH-adjusted culture versus controls that simply grew for 48 h or were subjected to all washes required for pH adjustment and then returned to the same supernatant (*Figure 3—figure supplement 1E*), possibly due to the high cell density and the transient nature of pH neutralization. We then assessed if increasing the pH at 30 h resulted in a decrease in the duration of lag phase by diluting the monoculture to OD = 0.0375 into fresh MRS after an additional 18 h of growth. Indeed, the lag phase was shorter for the pH-adjusted cells compared with a control culture (*Figure 3E*). Thus, pH is a driver of the growth advantages of *Lp* in lag phase even in the absence of a net increase in cell number.

## Co-culturing *Lp* with *Acetobacter* species reduces lag time

Canonical antibiotic tolerance in *E. coli* results from a decrease in growth rate or an increase in lag phase that protects cells through metabolic inactivity (*Brauner et al., 2016*). To measure growth rate and lag phase, we co-cultured *Lp* with each of the *Acetobacter* species individually for 48 hr, diluted the culture to a common OD of 0.0375, and monitored growth in a plate reader. The maximum growth rate was the same for the *Lp* monoculture and co-cultures with *Ap*, *At*, and *Ai*, and slightly higher for co-cultures with *Ao* and *Aa* (*Figure 4—figure supplement 1*). We previously observed for *Lp* monocultures when shifting the pH that the stimulation of growth in stationary phase was connected with a shorter lag phase (*Figure 3E*). In agreement with these data, there was a significant decrease in bulk lag time for all the *Acetobacter* co-cultures (*Figure 4A,B*). Surprisingly, this *shorter* lag phase was linked to *Lp* tolerance to rifampin, opposite to the longer phase linked to *E. coli* tolerance to ampicillin (*Brauner et al., 2016*). *Ap*, *At*, and *Ai* co-cultures had the largest lag decreases. The *Aa* and *Ao* co-cultures had smaller, although still significant, decreases (*Figure 4A, B*); interestingly, *Aa* and *Ao* were also less efficient at consuming lactate than the other *Acetobacter* species (*Figure 2C*). These data indicate that interspecies interactions can change the physiology of community members, and that differences across the *Acetobacter* species constitute an opportunity to probe the underlying cause of the lag phenotype.

As with *Lp* tolerance to antibiotics (*Figure 1B*), the shortened lag phase of the *Lp-Ap* co-culture depended on the inoculum coming from a co-culture. When we simply mixed independent 48 h cultures of *Lp* and *Ap*, the resulting bulk culture had the same lag time as an *Lp* monoculture (*Figure 4C,D*). To determine which of the two species was responsible for the decrease in lag, we performed time-lapse microscopy (Materials and methods) to monitor the initiation of growth at the single-cell level (*Figure 4E*). *Lp* and *Ap* were clearly distinguishable based on morphology (*Figure 4—figure supplement 2*): *Lp* cells are longer (2.46 ± 0.78 µm vs. 1.66 ± 0.38 µm) and thinner (0.72 ± 0.12 µm vs. 0.90 ± 0.09 µm) than *Ap* cells. Therefore, we used the aspect ratio (length/width; 3.41 ± 0.91 for *Lp* and 1.87 ± 0.46 for *Ap*) to distinguish single cells from each species in co-culture. We validated this strategy on co-cultures of fluorescently tagged strains of the same two species and measured a 10% error rate in classification (*Figure 4—figure supplement 2D–F*). In co-culture, most *Lp* cells were observed to have grown by 1 h after spotting, but in *Lp* monoculture, few cells were growing even after 2 h (*Figure 4E,F*). *Ap* cells did not grow during the time of imaging (*Figure 4F*), indicating that the reduced lag time was due to *Lp*'s growth, in agreement with CFU/mL measurements from liquid cultures (*Figure 2B*).

## Growth status and pH are drivers of antibiotic tolerance

Since pH changes shortened lag phase (*Figure 3E*) and changes in lag time were linked to antibiotic tolerance (*Figures 1* and *4*), we tested whether shortening lag phase was sufficient to induce tolerance. Increased time in starvation leads to changes in physiology, including an increase in the duration of lag phase (*Levin-Reisman et al., 2010*). To determine the relationship between time spent in stationary phase and lag time for *Lp*, we grew *Lp* cultures for varying amounts of time in stationary phase then used these cultures as inocula for growth in a plate reader. Incubating the *Lp*

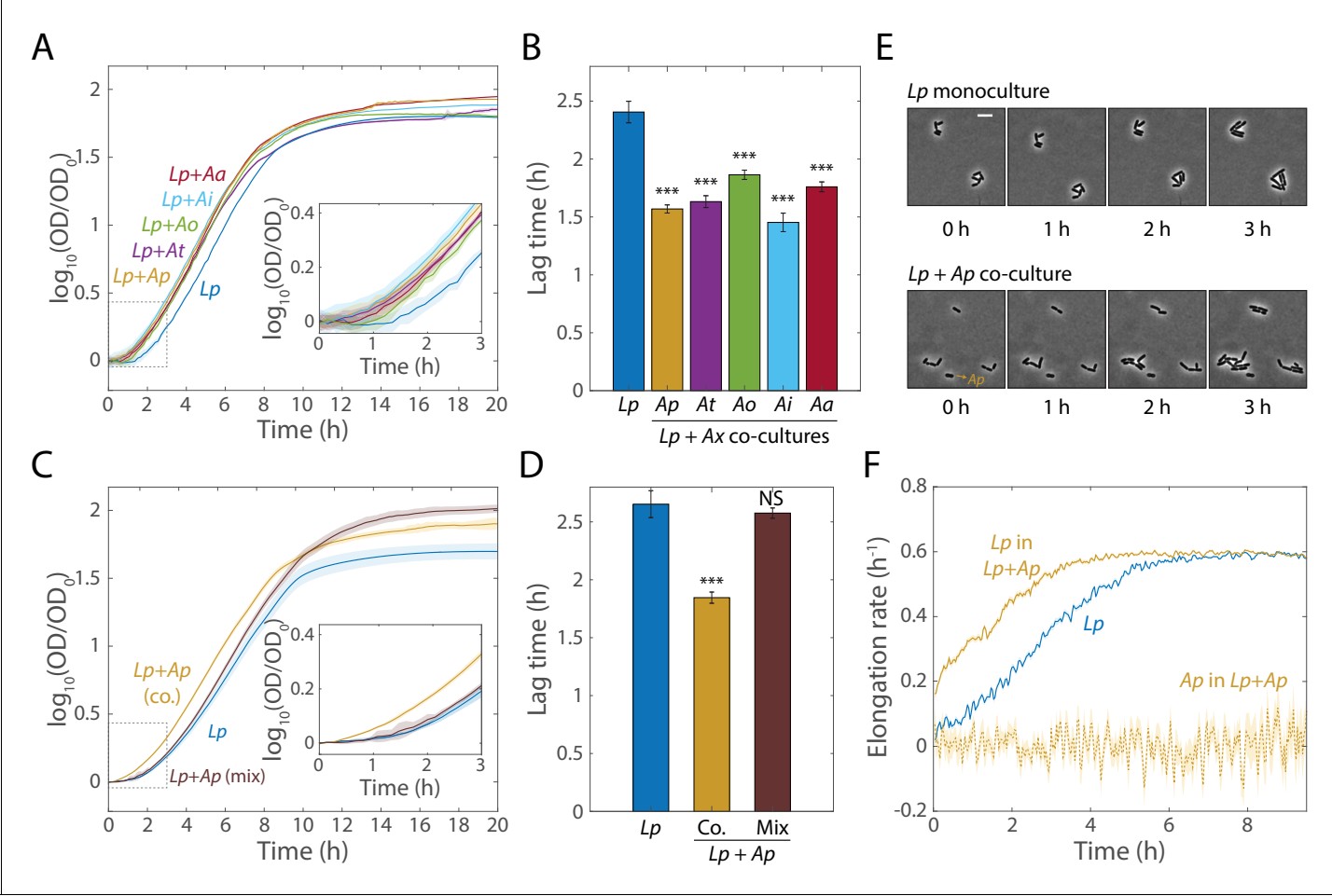

**Figure 4.** Co-cultures of *Lp* and *Acetobacter* species undergo shorter lag phases. (**A**) Calculating the logarithm of OD normalized by OD at $t$ = 0 reveals that co-cultures of *Lp* and various *Acetobacter* species (*Ax*) experience more rapid transitions from stationary phase to exponential growth than monocultures of *Lp*. Shaded regions indicate standard deviation (S.D.), $n$ = 5. Inset: zoom-in of region inside dashed box highlighting lag differences. Data points were taken every 8.3 min. (**B**) Co-culture lag times are significantly shorter than *Lp* monoculture lag times. Lag times were obtained by fitting the growth curves in (A) to the Gompertz equation. Error bars are S.D. for each condition, $n$ = 5. p-values are from a Student's two-sided $t$-test of the difference from the monoculture (***: $p < 2 \times 10^{-4}$, equivalent to ***: $p < 1 \times 10^{-3}/n$, where $n$ = 5 is the number of comparisons). (**C**) Mixing *Lp* monocultures with *Ap* monocultures (Mix) yields growth curves with a similar lag phase than those of *Lp* monocultures. Shaded regions indicate S.D., $n$ = 5. Inset: zoom-in on region inside dashed box highlighting lag differences. Data points were taken every 8.3 min. (**D**) Mixed *Lp*-*Ap* cultures do not experience significantly shorter lag times than *Lp* monocultures. Lag times were obtained by fitting the curves in (C) to the Gompertz equation. Error bars are S.D. for each condition, $n$ = 5. p-values are from a Student's two-sided $t$-test of the difference from the monoculture (***: $p < 5 \times 10^{-4}$, equivalent to ***: $p < 1 \times 10^{-3}/n$, where $n$ = 2 is the number of comparisons; NS: not significant). (**E**) Single-cell microscopy demonstrates that a decrease in the duration of lag phase of *Lp* was responsible for the lag-time decrease in co-culture. Representative phase microscopy images of *Lp* in monoculture and co-cultured with *Ap* on an MRS agar pad. The only *Ap* cell visible in these images is indicated with an arrow. Size bar = 5 μm. (**F**) The instantaneous elongation rate of single *Lp* cells increases faster in co-culture than in monoculture. Phase-contrast images were segmented and cells were classified as *Lp* or *Ap* based on their aspect ratio. Lines are the mean and shaded regions are the standard error for an *Lp* monoculture ($n_{Lp,0\ h}$ = 465, $n_{Lp,9.5\ h}$ = 27,503) or a co-culture with *Ap* ($n_{Lp,0\ h}$ = 448, $n_{Lp,9.5\ h}$ = 58,087, $n_{Ap,0\ h}$ = 47, $n_{Ap,9.5\ h}$ = 146). Images were taken every 5 min.

The online version of this article includes the following figure supplement(s) for figure 4:

**Figure supplement 1.** The growth rate of *L. plantarum* (*Lp*) co-cultures is similar to that of an *Lp* monoculture.

**Figure supplement 2.** Morphological differences permit differentiation of *L. plantarum* (*Lp*) from *Acetobacter pasteurianus* (*Ap*) in phase-contrast images.

monocultures for more than 48 h resulted in a dramatic increase in the duration of lag phase, while reducing the culturing time shortened lag phase (***Figure 5A***).

Because lag phase in co-culture is slightly shorter than that of a 24 h monoculture (***Figure 5A***), we decided to match the lag time of a co-culture by growing a monoculture for 20 h from an initial

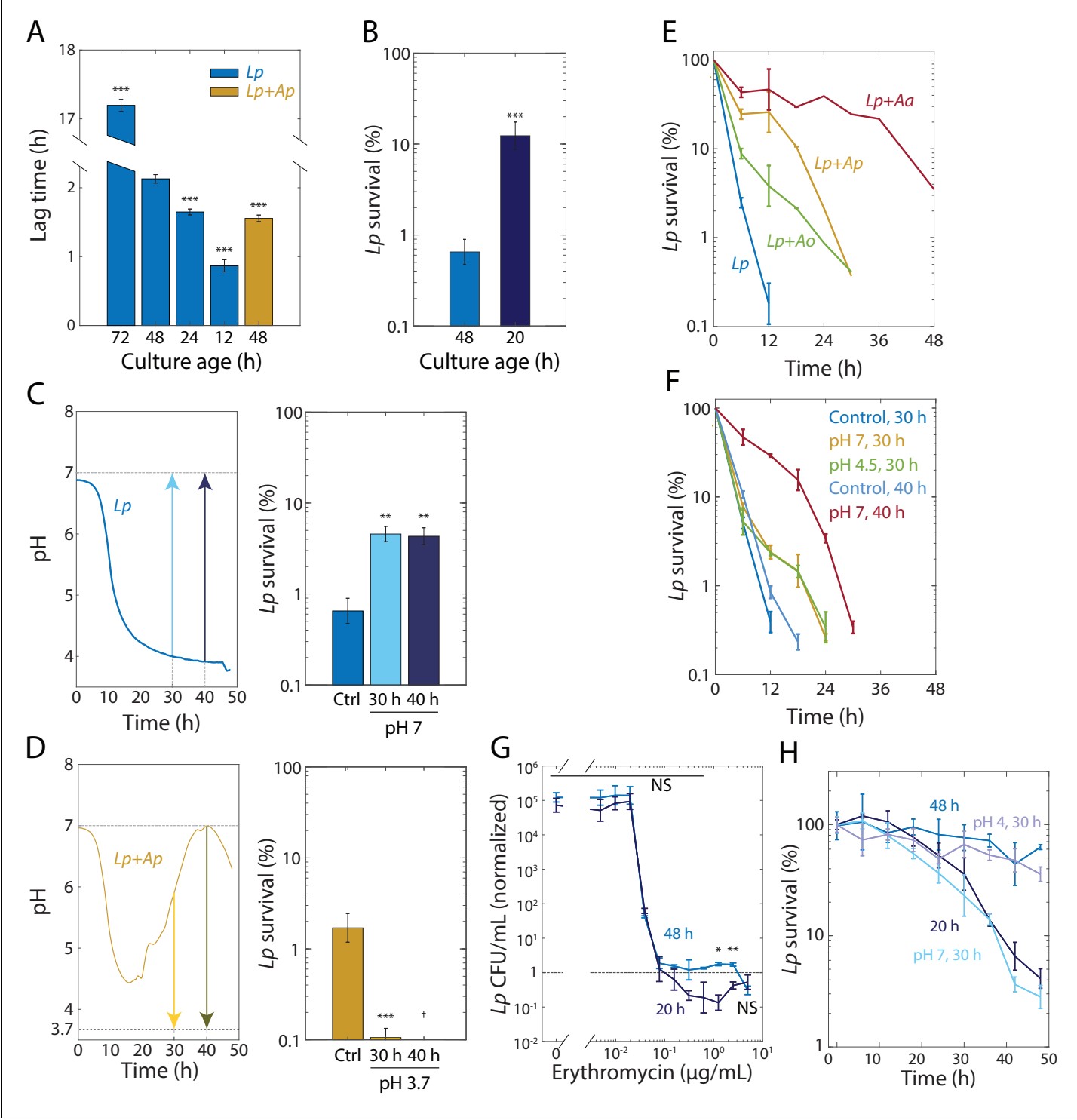

**Figure 5.** Tolerance to rifampin is modulated by pH. (**A**) The duration of lag phase of bulk cultures of *Lp* depends on the time spent in stationary phase. *Lp* monocultures grown for various times from OD = 0.02, and co-cultures with *Ap*, were diluted into fresh medium. Lag time was calculated by fitting growth curves to the Gompertz equation. Error bars are standard deviation (S.D.), $n = 12$. p-values are from a Student's two-sided *t*-test of the difference with respect to the 48 h culture (***: $p < 2.5 \times 10^{-4}$, equivalent to ***: $p < 1 \times 10^{-3}/n$, where $n = 4$ is the number of comparisons). (**B**) Culturing *Lp* as a monoculture for a shorter time leads to higher cell survival. Viable cell plating counts of *Lp* after growth in 20 µg/mL rifampin for 24 h normalized to the counts at the start of the experiment ($t = 0$). Error bars are S.D. for each condition, $n = 3$. p-values are from a Student's two-sided *t*-test of the difference between the cultures (***: $p < 1 \times 10^{-3}$). (**C**) Neutralization of pH in stationary phase in *Lp* monocultures is sufficient to induce tolerance. Increasing the pH of an *Lp* monoculture at $t = 30$ h or $t = 40$ h to 7 to mimic the pH increase in co-cultures of *Lp* with *Acetobacter* species

*Figure 5 continued on next page*

*Figure 5 continued*

(upper panel) increased cell survival (CFU/mL normalized to counts at the start of the experiment, $t = 0$) after treatment with 20 μg/mL rifampin for 24 h (lower panel). A 48-h-old culture with no changes in pH was used as a control (Ctrl.). Error bars are S.D. for each condition, $n = 3$. p-values are from a Student's two-sided $t$-test of the difference between the cultures (**: $p < 5 \times 10^{-3}$, equivalent to **: $p < 0.01/n$, where $n = 2$ is the number of comparisons). (D) Acidification of *Lp* co-cultures with *Ap* during the exponential-to-stationary phase transition or in late stationary phase sensitizes *Lp* to rifampin. Decreasing the pH of an *Lp* co-culture with *Ap* at $t = 30$ h or $t = 40$ h to 3.7 to mimic the pH of an *Lp* monoculture (upper panel) increased survival (CFU/mL normalized to counts at the start of the experiment, $t = 0$) after treatment with 20 μg/mL rifampin for 24 h (lower panel). Error bars are S.D. for each condition, $n = 3$. p-values are from a Student's two-sided $t$-test of the difference between the cultures (***: $p < 5 \times 10^{-4}$, equivalent to ***: $p < 1 \times 10^{-3}$, where $n = 2$ is the number of comparisons). †: values below the limit of detection. (E) The dynamics of killing in *Lp* co-culture with *Acetobacter* species differs quantitatively according to species and from *Lp* monoculture (blue), indicating that the *Acetobacter* species induce rifampin tolerance to different degrees. CFU/mL of *Lp* in monoculture and in co-culture with *Acetobacter* species, and treated with 50 μg/mL rifampin, normalized to counts at the start of the experiment ($t = 0$). Error bars are S.D. for each condition, $n = 3$. (F) The timing of the pH change in *Acetobacter* co-culture predicts the extent of protection against 50 μg/mL rifampin. Neutralization of pH in *Lp* monocultures at 40 h of growth (to mimic *Lp+Aa* co-cultures) elicits longer protection against rifampin than neutralization at 30 hr. A small increase in pH (from 3.85 to 4.5) at 30 h (to mimic *Lp+Ao* co-cultures) provides protection comparable to complete neutralization. CFU/mL were normalized to counts at the start of the experiment ($t = 0$). Error bars are S.D. for each condition, $n = 3$. (G) *Lp* survival to erythromycin is ~10 times higher after 24 h of treatment with erythromycin at supraMIC concentrations for 48-h-old monocultures of *Lp* than for 20-h-old *Lp* monocultures. Viable cell plating counts of *Lp* after growth in erythromycin for 24 h normalized to cell counts at the start of the experiment ($t = 0$). Error bars are S.D., $n = 3$. p-values are from a Student's two-sided $t$-test of the difference between the two samples at a given time point (*: $p < 4 \times 10^{-3}$, **: $p < 8 \times 10^{-4}$ equivalent to *: $p < 0.05/n$, **: $p < 0.01/n$, where $n = 12$ is the number of comparisons; NS: not significant). (H) Shifting the pH of an *Lp* monoculture at 30 h to 4 or 7, followed by 18 h of growth before treatment with 2 μg/mL erythromycin, mimics the survival dynamics of a 48-h-old or 20-h-old culture in stationary phase, respectively. CFU/mL of *Lp* monocultures were normalized to counts at the start of the experiment ($t = 0$). Error bars are S.D. for each condition, $n = 3$.

OD = 0.02. We then measured *Lp* survival in 20 μg/mL rifampin after 24 h in cultures diluted from a 48-hr-old or a 20-hr-old culture. Culturing for 20 h resulted in a significant increase in survival (*Figure 5B*). We next tested whether shortening lag phase by changing the pH in stationary phase also yielded increased rifampin tolerance of *Lp* in monoculture. We increased the pH of an *Lp* monoculture to 7 after 30 and 40 hr, and grew cells for an additional 18 h and 8 hr, respectively. We then measured the change in CFU/mL upon treatment with 50 μg/mL rifampin for 24 hr. The upshift in pH at $t = 30$ h or 40 h resulted in increased tolerance relative to the unshifted monoculture (*Figure 5C*). To test whether pH changes can also decrease tolerance, we grew co-cultures of *Lp* and *Ap* for a total of 48 hr, decreasing the pH to 3.7 at $t = 30$ h or 40 hr. In both cases, the viability after 24 h of rifampin exposure was significantly reduced relative to an untreated monoculture (*Figure 5D*). Thus, pH can affect tolerance both positively and negatively.

In co-culture with *Lp*, *Ap* raised the pH earlier than did *Aa*, while *Ao* only raised the pH very slightly (*Figure 2D*). We hypothesized that due to these distinct pH dynamics, rifampin would also have different *Lp* killing dynamics in these co-cultures. We grew co-cultures of these *Acetobacter* species with *Lp* as previously, and then treated the co-cultures with 50 μg/mL rifampin. While all co-cultures had extended survival relative to *Lp* monoculture, the killing dynamics of *Lp* were indeed distinct, with *Aa* inducing the highest tolerance (*Figure 5E*). To determine the extent to which these dynamics can be explained by the time at which each species raises the pH, we measured CFU/mL at various time points after rifampin treatment for *Lp* monocultures grown for 48 h whose pH was raised to pH seven at $t = 30$ or 40 hr, mimicking the early and late increases in pH for *Ap* and *Aa* co-cultures, respectively. The shift to pH 7 at 40 h induced higher rifampin tolerance than the shift at 30 h (*Figure 5F*), consistent with the increased tolerance of the *Lp-Aa* co-culture (*Figure 5E*). Moreover, shifting the pH to 4.5 at $t = 30$ hr, to mimic the slight increase caused by *Ao*, was also sufficient to increase tolerance comparable to pH neutralization at $t = 30$ h (*Figure 5F*), consistent with the similar killing dynamics of the *Ap* and *Ao* co-cultures (*Figure 5E*). All pH shifts induced higher tolerance compared to control cultures that underwent the same protocol but whose pH was maintained (*Figure 5F*). Taken together, these experiments establish that pH changes drive changes in both the exit from stationary phase and tolerance of *Lp* to rifampin.

## Growth status and pH also modulate tolerance to a ribosome-targeting antibiotic

The robust relationships among changes in pH, lag time, and rifampin tolerance prompted us to explore how changes in pH and lag time affect survival to other antibiotics. We decided to use the

ribosome-targeting macrolide erythromycin because it is bacteriostatic (in contrast to rifampin) and *Lp* is sensitive to it (*Supplementary file 1b*). We treated *Lp* monocultures grown for 20 h or 48 h with increasing concentrations of erythromycin for 24 h at a starting cell density of ~5 × 10$^5$ CFU/mL. Opposite to our observations with rifampin (*Figure 5B*), a 48 h *Lp* monoculture displayed tolerance to erythromycin, while a 20 h culture did not (*Figure 5G*). While both cultures had the same MIC in erythromycin (0.078 µg/mL, *Figure 5G*), at concentrations above the MIC, the 48 h culture showed no changes in CFU/mL after 24 h of erythromycin treatment; the 20 h culture had a reduction of ~10 fold in CFU/mL (*Figure 5G*). This result suggests that *Lp* cells are typically tolerant to erythromycin but killing is accelerated under conditions that make them tolerant to rifampin.

To determine whether antibiotic tolerance underlies the survival of *Lp* to erythromycin as well as to rifampin, we diluted 20- and 48 h cultures to a starting density of ~5 × 10$^5$ CFU/mL, exposed them to a high concentration of erythromycin (2 µg/mL, 25X MIC), and monitored CFU/mL over time. Cells from a 20 h culture died significantly more rapidly than cells from a 48-hr-old culture (*Figure 5H*), indicating that the differences in survival (*Figure 5G*) are explained by erythromycin tolerance. Further, increasing the pH of an *Lp* monoculture at 30 h and then exposing it after an additional 18 h of growth to 2 µg/mL erythromycin in fresh medium had an increase in the rate of killing that was similar to that achieved with a 20 h culture (*Figure 5H*). These results highlight that the effects of changing the growth status of a culture on antibiotic response are not limited to rifampin and—as in the case of erythromycin—can be opposite.

## Discussion

Our measurements of growth behavior in synthetic communities motivated by the natural context of the fly gut indicate that interspecies interactions impact both growth and the effect of antibiotics on individual species. The pH-based mechanism underlying the tolerance of *Lp* induced by *Acetobacter* species is intrinsically connected to the metabolic capacity of each species, and hence is likely to be generally relevant in more complex communities. Moreover, these findings could have important implications for human health, for example in the context of *Lactobacillus*-dominated vaginal microbiotas (*Boris and Barbés, 2000*), hence their generality should be tested broadly in other contexts.

In this study, we observed a novel form of antibiotic tolerance. Tolerance has been defined as increased time to killing of the population as a whole (*Brauner et al., 2017*), as opposed to resistance (a change in the MIC), or persistence or heterotolerance whereby a subpopulation of bacteria displays increased time to killing (*Balaban et al., 2019*; *Brauner et al., 2016*). *Lp* cells in monoculture rapidly died, while *Lp* cells in co-cultures showed close to 100% survival for extended periods (*Figures 1E* and *5E*). Furthermore, the time and magnitude of pH change affected the kinetics of killing but not the size of the surviving population (*Figure 5E and F*). These behaviors contrast with the stereotypical rapid decrease in viability, with a small subpopulation of surviving cells, observed in persistence. At high doses and late time points, cell counts were below our limit of detection and therefore we cannot rule out that, in addition to tolerance, pH changes in stationary phase could also modulate persistence. Tolerance to beta lactams such as ampicillin has been observed in *E. coli* cultures that exhibit slow growth or a long lag phase (*Brauner et al., 2016*), and *E. coli* mutants with longer lag phases can be selected through experimental evolution to match the time of treatment (*Fridman et al., 2014*; *Levin-Reisman et al., 2017*). Based on these previous studies, we were surprised to find the opposite effect with rifampin on *Lp*: cultures with a shorter lag phase exhibited increased tolerance (*Figures 1* and *4*). Moreover, although tolerance to erythromycin was associated with a longer lag phase (*Figures 3E* and *5A,H*), killing retardation was at least an order of magnitude longer than the change in lag time (*Figure 5A,H*), indicating that tolerance is not determined by an elongation of lag phase alone, in contrast to the effects of ampicillin on *E. coli* (*Fridman et al., 2014*). The opposite behaviors of *Lp* to rifampin and erythromycin upon pH changes suggest that tolerance is more complex than previously thought, and should motivate future investigations into the physiological and genetic basis of tolerance.

Several genetic factors that increase time to killing have been identified in *E. coli*, including toxin-antitoxin modules such as *hipBA* (*Rotem et al., 2010*) that induce the stringent response and thus cause transient growth arrest. In *Lp* co-culture with *Acetobacter* species, metabolic interactions alter the physiological state of *Lp* during late stationary phase by changing the environmental pH (*Figure 2*). The stringent response is required to survive acid shock in *Helicobacter pylori*

(*Mouery et al., 2006*) but not in *Enterococcus faecalis* (*Abranches et al., 2009*), which is in the same taxonomic order as *Lp*. In the case of *Lp*, whether the stringent response could be a major factor in the increased tolerance to rifampin is unclear due to the surprising connection with decreased lag.

The pH in stationary phase can affect many factors, such as the chemistry of extracellular metabolites and macromolecules as well as the surface of the cell (*Peters et al., 2016*). Importantly, our assays of antibiotic sensitivities were all performed at a starting pH of 7. Nonetheless, shifts in extracellular pH can lead to buffered drops in cytoplasmic pH (*Choi and Groisman, 2016*; *Olsen et al., 2002*) that can be regulated (*Chakraborty et al., 2015*) or result from internalization of low-p$K_a$ species such as short-chain fatty acids (*Ricke, 2003*). Such changes could lead to protonation of macromolecules involved in adsorption or changes in the proton motive force (*Krulwich et al., 2011*). How these factors affect non-polycationic antibiotics such as rifampin remains to be determined; neither of the ionizable functional groups of rifampin (p$K_a$s 1.7 and 7.9 *Maggi et al., 1966*) nor erythromycin (p$K_a$ 8.88 *McFarland et al., 1997*) have p$K_a$s in the pH range achieved in our cultures (*Figure 2*, *Figure 3—figure supplement 1B*). Protonation changes in target macromolecules could also lead to protection against antibiotics, although we would expect a subsequent change in MIC, contrary to our findings (*Figures 1* and *5*). Intracellular acidification by the short-chain fatty acid propionate has been proposed to lengthen lag phase in *Salmonella* in vitro and in the mouse gut (*Jacobson et al., 2018*), consistent with our finding that lag time (*Figure 4*) and intracellular pHluorin fluorescence (*Figure 2—figure supplement 3A*) are related.

Changes in intra- and extracellular pH have been shown to lead to transcriptional responses that provide cross-protection against antibiotics (*Komora et al., 2017*; *McMahon et al., 2007*; *O'Sullivan and Condon, 1997*), suggesting that the killing retardation due to a pH increase in stationary phase may result from a complex regulatory process. One major factor influencing the *Lactobacillus-Acetobacter* interaction is that these organisms form a recurrent community and may therefore have evolved to sense and benefit from each other's presence. Further experiments are needed to uncover the molecular mechanisms that link growth state and susceptibility to antibiotics in *Lactobacillus* species, other non-model organisms, and microbial communities. In addition, although we consistently observed related shifts in lag phase and tolerance (*Figures 3* and *5*), it remains to be established whether lag time and tolerance are causally linked or coupled to some global variable. The opposite connections with rifampin and erythromycin tolerance underscore the complexity of the link between growth and antibiotic action. Changes in growth may lead to changes in the levels and activity of these antibiotics' molecular targets. Moreover, the molecular mechanisms that lead to death downstream of the antibiotic target could be a function of the growth state of the cell. Previous work has shown that bacterial interactions can elicit changes in antibiotic sensitivity by changing cellular physiology or interfering with antibiotic action directly or indirectly (*Adamowicz et al., 2018*; *Radlinski et al., 2017*; *Sorg et al., 2016*).

In principle, a myriad of intra- and extracellular variables are subject to the composition and dynamics of the ecosystems that bacteria inhabit, and microbial communities within animal hosts can elicit changes in environmental variables both locally and globally. Specifically, the microaerobic and anaerobic microenvironments of the fly (*Obadia et al., 2017*) and human gastrointestinal tracts enable the growth of short chain fatty acid producers. Some of these short chain fatty acids, like butyrate, have been shown to play an important role on host physiology and health (*Nicholson et al., 2012*). The consequences of the accumulation of these short chain fatty acids and other small molecules on microenvironments, as well as their effect on bacterial physiology and antibiotic treatment efficacy in vivo, have yet to be systematically explored. Our findings linking short chain fatty acid metabolism, growth, and antibiotic action in commensal microbes from the fruit fly gut opens the door to studying these phenomena in a model organism. While the human gut microbiome comprises hundreds of bacterial species, the simplicity of the *Drosophila* gut microbiota (*Wong et al., 2011*), the genetic tractability of *Drosophila*, and the fact that ~ 65% of human disease-causing genes have homologs in the *Drosophila* genome (*Ugur et al., 2016*), make the fruit fly a powerful model for host-microbiome interactions (*Douglas, 2018*). Our results also emphasize the need to probe the action of antibiotics – as well as other drugs that are thought not to target microbial growth (*Maier et al., 2018*) – in complex and varied conditions (*Beppler et al., 2017*). Furthermore, our findings highlight the utility of studying growth physiology in co-cultures in the absence of antibiotics for uncovering novel mechanisms of community-encoded protection against antibiotics.

# Materials and methods

## Key resources table

| Reagent type (species) or resource | Designation | Source or reference | Identifiers | Additional information |
|---|---|---|---|---|
| Strain, strain background (*Drosophila melanogaster*) | *Wolbachia*-free *Drosophila melanogaster* Canton-S | Bloomington *Drosophila* Stock Center | BL64349 | |
| Strain, strain background (*Escherichia coli*) | BW29427 | Carol Gross lab | | *pir+* DAP- host strain *thrB1004, pro, thi, rpsL, hsdS, lacZDM15, RP4-1360 Δ(araBAD) 567 ΔdapA1341::[erm pir+]*, donor for conjugation with *Acetobacter pasteurianus* |
| Strain, strain background (*Lactobacillus plantarum*) | *Lp* | (*Obadia et al., 2017*) | | *Lactobacillus plantarum* (WF) wild fly (*D. melanogaster*) isolate |
| Strain, strain background (*Lactobacillus brevis*) | *Lb* | (*Obadia et al., 2017*) | | *Lactobacillus brevis* lab fly (Oregon-R) isolate |
| Strain, strain background (*Acetobacter pasteurianus*) | *Ap* | (*Gould et al., 2018*) | | *Acetobacter pasteurianus* lab fly (Oregon-R) isolate |
| Strain, strain background (*Acetobacter tropicalis*) | *At* | (*Gould et al., 2018*) | | *Acetobacter tropicalis* lab fly (Oregon-R) isolate |
| Strain, strain background (*Acetobacter orientalis*) | *Ao* | (*Obadia et al., 2017*) | | *Acetobacter orientalis* lab fly (Oregon-R) isolate |
| Strain, strain background (*Acetobacter indonesiensis*) | *Ai* | (*Obadia et al., 2017*) | | *Acetobacter indonesiensis* lab fly isolate |
| Strain, strain background (*Acetobacter aceti*) | *Aa* | (*Obadia et al., 2017*) | | *Acetobacter aceti* lab fly isolate |
| Strain, strain background (*Lactobacillus plantarum*) | *Lp* mCherry | (*Obadia et al., 2017*) | | *Lactobacillus plantarum* (WF) wild fly isolate pCD256-P11-mCherry |
| Strain, strain background (*Lactobacillus plantarum*) | $Lp^{pH}$ | This study | | *Lactobacillus plantarum* (WF) wild fly isolate pCD256-P11-pHluorin |
| Strain, strain background (*Acetobacter pasteurianus*) | *Ap* GFP | This study | | *Acetobacter pasteurianus* lab fly (Oregon-R) isolate pCM62-Plac-sfGFP |
| Recombinant DNA reagent | pCM62 (plasmid) | (*Marx and Lidstrom, 2001*) | | Plasmid to clone sfGFP under the control of the *Escherichia coli* lac promoter |

*Continued on next page*

*Continued*

| Reagent type (species) or resource | Designation | Source or reference | Identifiers | Additional information |
|---|---|---|---|---|
| Recombinant DNA reagent | pCD256-mCherry (plasmid) | (*Spath et al., 2012a*) | | Backbone for pHluorin expression |
| Recombinant DNA reagent | pBad-sfGFP (plasmid) | Addgene | RRID: Addgene_85482 | Source of sfGFP |
| Recombinant DNA reagent | pZS11-pHluorin | (*Mitosch et al., 2017*) | | Source of pHluorin |
| Sequence-based reagent | ZTG109 | This study | PCR primers | ggatttatgcATGAG CAAGGGCGAGGAG |
| Sequence-based reagent | ZTG110 | This study | PCR primers | gctttgttagcagccgga tcgggcccggatctcgag TTACTTGTACAGC TCGTCCATG |
| Sequence-based reagent | ZFH064-pHluorin | This study | PCR primers | ATTACAAGGAGAT TTTACAT ATGAGTA AAGGAGAAGAAC TTTTC |
| Sequence-based reagent | ZFH065-pHluorin | This study | PCR primers | gtctcggacagcggttttG GATCCTTATTTGTATA GTTCATCCATG |
| Commercial assay or kit | Cell Viability Kit | BD | 349483 | |
| Commercial assay or kit | EnzyChrom L-lactate Assay Kit | BioAssay Systems | ECLC-100, Lots BH06A30 and BI07A09 | |
| Commercial assay or kit | EnzyChrom D-lactate Assay Kit | BioAssay Systems | EDLC-100, Lots BH0420 and BI09A07 | |
| Chemical compound, drug | D-mannitol | ACROS Organics | AC125345000, Lot A0292699 | |
| Chemical compound, drug | lactate | Sigma | L6661-100ML Lot MKCC6092 | |
| Chemical compound, drug | Tween 80 | ACROS Organics | AC278632500 Lot A0375189 | |
| Chemical compound, drug | NaOH | EMD Millipore | SX0590, Lot B0484969043 | |
| Chemical compound, drug | HCl | Fisher Chemical | A144-500, Lot 166315 | |
| Chemical compound, drug | ampicillin | MP Biomedicals | 02194526, Lot R25707 | |
| Chemical compound, drug | streptomycin | Sigma | S9137 Lot SLBN3225V | |
| Chemical compound, drug | chloramphenicol | Calbiochem | 220551, Lot D00083225 | |
| Chemical compound, drug | tetracycline | MP Biomedicals | 02103011, Lot 2297K | |

*Continued on next page*

*Continued*

| Reagent type (species) or resource | Designation | Source or reference | Identifiers | Additional information |
|---|---|---|---|---|
| Chemical compound, drug | erythromycin | Sigma | E5389-1G, Lot WXBC4044V | |
| Chemical compound, drug | ciprofloxacin | Sigma-Aldrich | 17850, Lot 116M4062CV | |
| Chemical compound, drug | trimethoprim | Alfa Aesar | J63053-03, Lot T16A009 | |
| Chemical compound, drug | spectinomycin | Sigma-Aldrich | PHR1426-500MG, Lot LRAA9208 | |
| Chemical compound, drug | rifampin | Sigma | R3501-5G, Lot SLBP9440V | |
| Chemical compound, drug | vancomycin | Sigma-Aldrich | PHR1732—4 × 250 MG, Lot LRAB3620 | |
| Chemical compound, drug | BCECF | Invitrogen | B1151, Lot 1831845 | |
| Chemical compound, drug | DMSO | Fisher BioReagents | BP231, Lot 165487 | |
| Software, algorithm | MATLAB | Mathworks | RRID:SCR_01622 | R2018a |
| Software, algorithm | μManager | (*Edelstein et al., 2010*) | RRID:SCR_016865 | v. 1.4 |
| Software, algorithm | Morphometrics | (*Ursell et al., 2017*) | | |
| Software, algorithm | SuperSegger | (*Stylianidou et al., 2016*) | | v. 3 |
| Other | MRS medium | BD | 288110 | |
| Other | yeast extract | Research Products International | Y20020, Lot 30553 | |
| Other | peptone | BD | 211677 Lot 7065816 | |
| Other | agar | BD | 214530 | |
| Other | PBS | Gibco | 70011044 | (10X, pH 7.4) |

## Fruit fly stocks and gut microbiome sequencing

*Wolbachia*-free *Drosophila melanogaster* Canton-S (BL64349) flies were obtained from the Bloomington *Drosophila* Stock Center, and were reared and maintained as previously described in *Obadia et al. (2017)*. To determine the bacterial strains present in our flies, we performed culture-independent 16S amplicon sequencing targeting the V4 region on an Illumina MiSeq. Individual flies were $CO_2$-anesthetized, surface-sterilized by washing with 70% ethanol and sterile PBS six times each. Flies were dissected under a stereo microscope and their guts were placed in 2 mL screw cap microtubes containing 200 μL of 0.1 mm sterile zirconia-silicate beads (BioSpec Products 11079101z) and 350 μL of sterile lysis buffer (10 mM Tris-HCl, pH 8, 25 mM NaCl, 1 mM EDTA, 20 mg/mL lysozyme). Samples were homogenized by bead beating at maximum speed (Mini-Beadbeater, BioSpec Products) for 1 min. Proteinase K was added at 400 μg/mL and samples were incubated for 1 h at 37˚C. Sample were then centrifuged (3000 × *g* for 3 min) and 300 μL of the nucleic acids-containing supernatant were transferred to 1.7 mL microtubes. Genomic DNA from samples was cleaned up

through a DNA Clean and Concentrator-5 column (Zymo Research D4014). Using the protocol described in *Fadrosh et al. (2014)* for library preparation and sequencing, we sequenced the gut contents of 18 individual flies, three flies each from six independent vials. Paired-end 250-base pair sequencing generated >10,000 reads per sample. Reads were filtered using PrinSeq as in *Koch et al. (2016)*. The reads were then clustered into operational taxonomic units (OTUs) at 99% identity and assigned taxonomy using LOTUS (*Hildebrand et al., 2014*) with the following parameters: [-threads 60 -refDB SLV -highmem 1 -id 0.99 p miseq -useBestBlastHitOnly 1 -derepMin 3:10,10:3 -simBasedTaxo 1 CL 3]. Redundant strain identities were collapsed into single OTUs. Common reagent contaminant strains were then removed (*Salter et al., 2014*). After filtering, only five unique species were identified (*Figure 1A*). We isolated these species in culture and verified the taxonomic identity of our isolates using Sanger sequencing of the complete 16S rRNA gene (*Gould et al., 2018*). At 97% OTU clustering, only three species were found: *Acetobacter* sp., *Lactobacillus plantarum*, and *Lactobacillus brevis*. When less stringent FASTQ quality filtering was used, trace amounts (~0.01%) of two mammalian gut strains were identified: *Blautia* sp. and *Bacteroides* sp. Because these OTUs were eliminated by more stringent quality filtering, we speculate that they may have resulted from barcode bleed-through on the MiSeq flowcell.

Sequencing data is available at the NCBI website under BioProject accession number PRJNA530819.

## Bacterial growth and media

Bacterial strains used in this study are listed in *Supplementary file 1a*. For culturing, all strains were grown in MRS medium (Difco Lactobacilli MRS Broth, BD 288110). MYPL medium was adapted from *Moens et al. (2014)*, with 1% (w/v) D-mannitol (ACROS Organics AC125345000, Lot A0292699), 1% (w/v) yeast extract (Research Products International Y20020, Lot 30553), 0.5% (w/v) peptone (Bacto-peptone, BD 211677 Lot 7065816), 1% (w/v) lactate (Lactic acid, Sigma L6661-100ML Lot MKCC6092), and 0.1% (v/v) Tween 80 (Polyoxyethylene(20)sorbitan monooleate, ACROS Organics AC278632500 Lot A0375189). The medium was set to pH 7 with NaOH (EMD Millipore SX0590, Lot B0484969043). All media were filter-sterilized.

Frozen stocks were streaked onto MRS agar plates (1.5% agar, Difco agar, granulated, BD 214530) and single colonies were picked to start cultures. Colonies were inoculated into 3 mL MRS in glass test tubes and grown for 48 h at 30℃ with constant shaking. Unless otherwise noted, the saturated cultures were diluted to OD = 0.02 and grown in 3 mL MRS in glass test tubes for 48 h at 30℃ with constant shaking before the start of the experiment. For co-cultures, we grew *Lp* and each of the *Acetobacter* species separately for 48 h from colonies as above, diluted the monocultures to OD = 0.04, combined the *Lp* monoculture 1:1 with each *Acetobacter* monoculture, and grew them in 3 mL MRS in glass test tubes for 48 h at 30℃ with constant shaking before the start of the experiment. The experiments shown in the Figures were conducted at 30℃ in either: 200 μL in 96-well polypropylene plates with constant shaking in a plate reader (*Figure 1B*, *Figure 1—figure supplement 1*, *Figure 1—figure supplement 2*, *Figure 2C,D*, *Figure 2—figure supplement 1A*, *Figure 2—figure supplement 2B*, *Figure 3A–D*, *Figure 3—figure supplement 1A–D*, *Figure 4A–D*, *Figure 4—figure supplement 1*, and *Figure 5G–H*); 3 mL in glass test tubes with constant shaking (*Figure 1C–E*, *Figure 2A,B*, *Figure 2—figure supplement 1*, *Figure 2—figure supplement 2A*, *Figure 2—figure supplement 3*, *Figure 3E*, *Figure 3—figure supplement 1E*, and *Figure 5A–F*); or on MRS agarose pads (see below, *Figure 1—figure supplement 2D*, *Figure 4E,F*, and *Figure 4—figure supplement 2*).

To count CFUs in cultures, aliquots were diluted serially in PBS. For cultures treated with high concentrations of antibiotics, cells were centrifuged for 1.5 min at 8000 x *g* and resuspended in 1X PBS pH 7.4 (Gibco 70011044) after removing the supernatants. PBS-diluted cultures were plated on MRS and MYPL because *Lactobacillus* species grow faster than *Acetobacter* species on MRS and vice versa on MYPL. Colony morphology and color enable differentiation of *Lactobacillus* from *Acetobacter* species.

## Conditioned media

Conditioned media were obtained by centrifuging cultures at 4500 x *g* for 5 min and filtering the supernatant with a 0.22 μm polyethersulfone filter (Millex-GP SLGP033RS) to remove cells.

Conditioned media were acidified with HCl (Fisher Chemical A144-500, Lot 166315) or basified with NaOH (EMD Millipore SX0590, Lot B0484969043). Conditioned media were sterilized after adjusting pH with 0.22 μm PES filters.

## MIC estimations

To estimate the sensitivity of each species to various antibiotics, colonies were inoculated into MRS and grown for 48 h at 30°C with constant shaking. Cultures were diluted to an OD of 0.001 for *Lp*, *Lb*, and *At*, and 0.01 for *Ap*. Diluted cultures (195 μL) were transferred to 96-well plates containing 5 μL of antibiotics at 40X the indicated concentration. Antibiotics used were ampicillin (ampicillin sodium salt, MP Biomedicals 02194526, Lot R25707, stock at 100 mg/mL in milliQ $H_2O$), streptomycin (streptomycin sulfate salt, Sigma S9137 Lot SLBN3225V, stock at 50 mg/mL in milliQ $H_2O$), chloramphenicol (Calbiochem 220551, Lot D00083225, stock at 50 mg/mL in ethanol), tetracycline (tetracycline hydrochloride, MP Biomedicals 02103011, Lot 2297K, stock at 25 mg/mL in dimethyl sulfoxide (DMSO)), erythromycin (Sigma E5389-1G, Lot WXBC4044V, stock at 64 mg/mL in methanol), ciprofloxacin (Sigma-Aldrich 17850, Lot 116M4062CV, stock at 1.2 mg/mL in DMSO), trimethoprim (Alfa Aesar J63053-03, Lot T16A009, stock at 2 mg/mL in DMSO), spectinomycin (spectinomycin hydrochloride, Sigma-Aldrich PHR1426-500MG, Lot LRAA9208, stock at 50 mg/mL in milliQ $H_2O$), rifampin (Sigma R3501-5G, Lot SLBP9440V, stock at 50 mg/mL in DMSO), and vancomycin (vancomycin hydrochloride, Sigma-Aldrich PHR1732−4 × 250 MG, Lot LRAB3620, stock at 200 mg/mL in DMSO:$H_2O$ 1:1). Antibiotics were diluted serially in 2-fold increments into MRS. Cultures were grown for 24 h at 30°C with constant shaking and absorbance was measured in an Epoch2 plate reader (BioTek Instruments) at 600 nm. The MIC was estimated as the minimum concentration of antibiotic with absorbance within two standard deviations of media controls.

For experiments in *Figure 1B* and *Figure 1—figure supplement 2A–C*, mono- and co-cultures were diluted to an OD of 0.001 (final cell density ~5 × $10^5$ CFU/mL) and transferred to 96-well plates containing 5 μL of rifampin at 40X working concentration. Cultures were grown for 24 h and then serially diluted in 5-fold increments in PBS, and 3 μL of the dilutions were spotted onto MRS and MYPL rectangular plates using a semi-automated high-throughput pipetting system (BenchSmart 96, Mettler Toledo). Plates were incubated at 30°C until colonies were visible for quantification of viability.

## Plate reader growth curves

Cultures were grown from single colonies for 48 h in MRS at 30°C with constant shaking. Then, cultures were diluted to a final OD of 0.02 and 200 μL of the dilutions were transferred to clear-bottom transparent 96-well plates. Plates were sealed with transparent film pierced with a laser cutter to have ~0.5 mm holes to allow aeration in each well. Absorbance was measured at 600 nm in an Epoch2 plate reader (BioTek Instruments). Plates were shaken between readings with linear and orbital modes for 145 s each.

Growth rates and lag times were quantified using MATLAB (Mathworks, R2008a). The natural logarithm of OD was smoothed with a mean filter with window size of 5 timepoints for each condition over time, and the smoothed data were used to calculate the instantaneous growth rate $d(\ln(OD))/dt$. The smoothed ln(OD) curve was fit to the Gompertz equation (*Zwietering et al., 1990*) to determine lag time and maximum growth rate.

## pH measurements

Culture pH was measured using the dual-excitation ratiometric pH indicator 2',7-bis-(2-carboxyethyl)−5-(and-6)-carboxyfluorescein, mixed isomers (BCECF, Invitrogen B1151, Lot 1831845), which has a p$K_a$ of ~6.98. A stock solution of 1 mg/mL BCECF in DMSO (Fisher BioReagents BP231, Lot 165487) was diluted 1000-fold into MRS to a final concentration of 1 μg/mL. Cells were grown in a Synergy H1 plate reader (BioTek Instruments) following the procedure described above. In addition to absorbance, fluorescence was measured every cycle using monochromators at excitation (nm)/emission (nm) wavelength combinations 440/535 and 490/535. After subtracting the fluorescence of wells containing cells without the indicator, the ratio of the signals excited at 490 nm and 440 nm was used to calculate the culture pH using a calibration curve of MRS set to various pH values.

Culture pH after 48 h of growth was directly measured with a pH meter (sympHony, VWR) equipped with a pH combination electrode (Fisherbrand accumet 13-610-104A).

## Changes in pH during growth

To change the pH of monocultures and co-cultures in stationary phase, we obtained conditioned medium at 30 h or 40 h as described above and set the pH to the desired values. We then centrifuged 2 mL of a replicate culture for 3 min at 8000 x $g$, removed the supernatant, and resuspended cells in 1 mL of the corresponding medium to wash the cells. The suspension was centrifuged a second time and the pellets were resuspended in 2 mL of the corresponding medium.

## Time-lapse and fluorescence microscopy

Cells were imaged on a Nikon Eclipse Ti-E inverted fluorescence microscope with a 100X (NA 1.40) oil-immersion objective. Images were collected on a DU897 electron multiplying charged couple device camera (Andor) using µManager v. 1.4 (*Edelstein et al., 2010*). Cells were maintained at 30°C during imaging with an active-control environmental chamber (Haison).

Cultures grown for 48 h were diluted 100-fold into PBS and 2 µL were spotted onto a 1% (w/v) agarose MRS pad. After drying at room temperature, the pads were covered with a cover slip, sealed with a mixture of equal portions of Vaseline, lanolin, and paraffin, and transferred to the microscope. Images were taken every 2 min using µManager v. 1.4.

To quantify the morphology of cells using fluorescent strains, co-cultures were diluted 100-fold into PBS and 2 µL were spotted onto a 1% (w/v) agarose PBS pad. After drying, the pads were covered with a cover slip and transferred to the microscope. Images were acquired at room temperature using µManager v. 1.4.

## Single-cell tracking and analysis

Images were segmented and cells were tracked using the software *SuperSegger* v. 3 (*Stylianidou et al., 2016*). Further analysis of single-cell growth was performed using MATLAB. Cells with length >6 µm were removed from further analysis due to issues with segmentation. Length traces were smoothed using a mean filter of window size 5. Cells were classified as *Lp* or *Ap* if 90% of their traces were above (*Lp*) or below (*Ap*) a $\log_{10}$(length-to-width ratio) of 0.375. Traces with more than 15 timepoints were used for further analysis. Elongation rates $d(\ln L)/dt$ were calculated for each cell and the mean and standard error were computed for each time point.

## Cloning and transformations

To generate the fluorescently labeled *Ap* strain, the *sfGFP* coding sequence was cloned into pCM62 (*Marx and Lidstrom, 2001*) under control of the *Escherichia coli* lac promoter. The sfGFP coding sequence was amplified from pBAD-sfGFP using primers ZTG109 (5' ggatttatgcATGAGCAAGGGC-GAGGAG) and ZTG110 (5'- gctttgttagcagccggatcgggcccggatctcgagTTACTTGTACAGCTCGTCCATG). Gibson assembly (*Gibson et al., 2009*) was used to insert the amplified *sfGFP* cassette into BglII/XhoI-digested pCM62. This construct was delivered into *Ap* by conjugation as previously described (*Deeraksa et al., 2005*). *Escherichia coli* BW29427 was used as a donor strain and maintained with 80 mg/mL 2,6-diaminopimelic acid (Sigma Aldrich 33240) in potato agar mating plates (*Deeraksa et al., 2005*). Transformed *Ap* was selected with 10 µg/mL tetracycline on yeast peptone glycerol agar plates (*Deeraksa et al., 2005*).

To generate the *Lp* strain harboring pHluorin, the pHluorin coding sequence was cloned into pCD256-mCherry (*Spath et al., 2012a*) under the control of the strong p11 promoter (*Rud et al., 2006*). The pHluorin coding sequence was amplified from pZS11-pHluorin (*Mitosch et al., 2017*) using primers ZFH064-pHluorin (5'-ATTACAAGGAGATTTTACAT ATGAGTAAAGGAGAAGAAC TTTTC) and ZFH065-pHluorin (5'-gtctcggacagcggttttGGATCCTTATTTGTATAGTTCATCCATG). Gibson assembly (*Gibson et al., 2009*) was used to insert the amplified pHluorin cassette into NdeI/BamHI-digested pCD256-mCherry. The *Lp*-pHluorin strain was generated by transforming wild type *Lp* as previously described (*Spath et al., 2012b*).

Fluorescent strains were further grown in MRS with antibiotics (10 µg/mL chloramphenicol (Calbiochem 220551, Lot D00083225) for *Lp*, tetracycline (10 µg/mL tetracycline hydrochloride, MP Biomedicals 02103011, Lot 2297K) for *Ap*).

## pHluorin measurements

Cells were grown following the procedure described above. The *Lp* pHluorin strain was grown in MRS containing 10 μg/mL chloramphenicol for the first 48 h of growth. In addition to absorbance, fluorescence was measured every cycle using monochromators at excitation (nm)/emission (nm) wavelength combinations 405/509 and 475/509. Because the signal from excitation wavelength 405 nm was indistinguishable from signal from medium (data not shown), we also measured pHluorin signal at both excitation/emission wavelength combinations for cells in PBS. Cultures (48-h-old, 250 μL) were centrifuged at 10,000 x *g* for 1 min and resuspended in 1X PBS. Aliquots (200 μL) were transferred to a 96-well plate and fluorescence was measured using monochromators at excitation (nm)/emission (nm) wavelength combinations 405/509 and 475/509 within 1 min of resuspension in a Synergy H1 plate reader (BioTek Instruments).

## Lactate measurements

Colonies of *Lp* and *Acetobacter* species were inoculated into 3 mL MRS and grown for 48 h at 30°C with constant shaking. Saturated cultures were diluted to OD 0.02, mixed 1:1, and grown at 30°C with constant shaking. After mixing for 20 h and 48 hr, a 700 μL aliquot was transferred to a microcentrifuge tube and centrifuged at 10,000 x *g* for 4 min. Supernatant (600 μL) was transferred to a new tube and centrifuged at 10,000 x *g* for 4 min. Supernatant (500 μL) was transferred to a new tube and kept on ice for not longer than 1 hr, until lactate was measured.

L- and D-lactate concentrations were measured using the EnzyChrom L- (BioAssay Systems ECLC-100, Lots BH06A30 and BI07A09) and D-lactate (BioAssay Systems EDLC-100, Lots BH0420 and BI09A07) Assay Kits. Samples were diluted 10- and 100-fold in water, and absorbance was measured according to the manufacturer's instructions in a plate reader (Tecan M200). We also included controls without lactate dehydrogenase to account for endogenous activity in the supernatants.

## Statistical analyses

To determine significance of differences, we performed pairwise Student's two-sided *t*-tests throughout. To decrease Type I error, we performed Bonferroni corrections for each experiment. Significant differences are denoted in the figures: *: $p<0.05/n$, **: $p<0.01/n$, ***: $p<0.001/n$, where $n$ is the number of comparisons.

The number of technical replicates are indicated in the figure legends. In all cases, technical replicates originated from a single bacterial colony, which was grown for 48 h in liquid media. The saturated culture was then split into technical replicates. Biological replicates correspond to experiments starting from a different bacterial colony, freshly streaked from frozen stocks. Biological replicates of the main conclusions of this study are present in multiple figures: co-culturing of *Lp* with *Ap* protects *Lp* from rifampin killing (*Figure 1B,C,D*); the protective effect is due to a change in killing kinetics (*Figure 1E* and *Figure 5E*); co-culturing of *Lp* with *Ap* leads to an increase in pH in stationary phase (*Figure 2D* and *Figure 2—figure supplement 2B*); and co-culturing of *Lp* with *Ap* shortens lag phase (*Figure 4A,C* and *Figure 5A*).

## Acknowledgements

The authors thank Vivian Zhang for technical support, Elizabeth Skovran for kindly providing the pCM62 plasmid for *Acetobacter* spp., and Kazunobu Matsushita for providing the *A tropicalis* SKU1100 control strain in the initial conjugation experiments. We also thank the Huang and Ludington labs for fruitful discussions. This work was supported by NIH Director's New Innovator Awards DP2OD006466 (to KCH), NSF CAREER Award MCB-1149328 (to KCH), the Allen Center for Systems Modeling of Infection (to KCH), and NIH Director's Early Independence Award DP5OD017851 (to WBL). KCH is a Chan Zuckerberg Biohub Investigator. AA-D is a Howard Hughes Medical Institute International Student Research fellow and a Stanford Bio-X Bowes fellow.

## Additional information

### Funding

| Funder | Grant reference number | Author |
|---|---|---|
| National Institutes of Health | DP2OD006466 | Kerwyn Casey Huang |
| National Science Foundation | MCB-1149328 | Kerwyn Casey Huang |
| Paul G. Allen Family Foundation | | Kerwyn Casey Huang |
| National Institutes of Health | DP5OD017851 | William B Ludington |
| Howard Hughes Medical Institute | International Student Research Fellowship | Andrés Aranda-Díaz |
| Chan Zuckerberg Biohub | | Kerwyn Casey Huang |
| Stanford University | Bio-X Graduate Fellowship | Andrés Aranda-Díaz |

The funders had no role in study design, data collection and interpretation, or the decision to submit the work for publication.

### Author contributions

Andrés Aranda-Díaz, Conceptualization, Resources, Data curation, Software, Formal analysis, Supervision, Validation, Investigation, Visualization, Methodology, Project administration; Benjamin Obadia, Ren Dodge, Data curation, Formal analysis, Investigation, Methodology; Tani Thomsen, Data curation, Formal analysis, Investigation; Zachary F Hallberg, Resources, Investigation, Methodology; Zehra Tüzün Güvener, Resources, Investigation; William B Ludington, Kerwyn Casey Huang, Conceptualization, Resources, Formal analysis, Supervision, Funding acquisition, Visualization, Methodology, Project administration

### Author ORCIDs

Andrés Aranda-Díaz https://orcid.org/0000-0002-0566-4901
Benjamin Obadia http://orcid.org/0000-0002-3286-3236
William B Ludington https://orcid.org/0000-0001-9637-4493
Kerwyn Casey Huang https://orcid.org/0000-0002-8043-8138

### Decision letter and Author response

Decision letter https://doi.org/10.7554/eLife.51493.sa1
Author response https://doi.org/10.7554/eLife.51493.sa2

## Additional files

### Supplementary files

• Supplementary file 1. Strains used in this study and their minimum inhibitory concentration. A) Strains were isolated from wild or laboratory *D. melanogaster* and were identified via plating assays by their colony morphologies. B) Minimum inhibitory concentrations expressed in µg/mL ± standard deviation, *n* = 3.

• Transparent reporting form

### Data availability

All data generated or analyzed during this study are included in the manuscript and supporting files, excepting sequencing data that have been deposited in the sequence read archive of NCBI under accession number PRJNA530819 (https://www.ncbi.nlm.nih.gov/bioproject/PRJNA530819/).

The following dataset was generated:

| Author(s) | Year | Dataset title | Dataset URL | Database and Identifier |
|-----------|------|---------------|-------------|-------------------------|
| Ludington W | 2019 | Laboratory *Drosophila melanogaster* gut microbiome | https://www.ncbi.nlm.nih.gov/bioproject/PRJNA530819/ | NCBI BioProject, PRJNA530819 |

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
