## [Decision Letter]

**Acceptance summary:**

Aranda-Diaz et al., describe how microbial interactions alter the impact of antibiotics on bacteria found in the gut of fruit flies. The authors find that a strain of *Lactobacillus plantarum* (Lp) is killed more slowly by rifampin when the bacteria are co-cultured with *Acetobacter* species also found in fly guts. In this system, *L. plantarum* increases the pH of growth media through the release of lactate. *Acetobacter* reduce acidification and reduce the lag time of *L. plantarum* in co-culture. It appears that the reduced impact of rifampin on *L. plantarum* in co-culture is driven by the impact that *Acetobacter* have on pH and lagtime. The reduced killing can be recapitulated by either altering the pH or the lagtime of *L. plantarum* in monoculture. This work adds to the growing literature demonstrating that bacterial interactions can alter the impact of antibiotics. The described mechanism of increasing tolerance by reducing lag time is novel.

**Decision letter after peer review:**

Thank you for submitting your article "Bacterial interspecies interactions modulate pH-mediated antibiotic tolerance" for consideration by *eLife*. Your article has been reviewed by 4 peer reviewers and the evaluation was overseen by Wenying Shou as the Reviewing Editor and Wendy Garrett as the Senior Editor. The following individuals involved in review of your submission have agreed to reveal their identity: Jianxiao Song (Reviewer #2); Jeff Gore (Reviewer #4).

The reviewers have discussed the reviews with one another and the Reviewing Editor has drafted this decision to help you prepare a revised submission.

Summary:

Understanding how antibiotics impact multi-species microbial communities is an important challenge to microbiology and medicine. In this study, the authors find that antibiotic tolerance of *Lactobacillus plantarum (Lp*) to rifampicin increases in the presence of other members of the fruit fly microbiome, and that this increase in tolerance is mediated by pH changes of the media induced by microbial metabolism. Surprisingly, the increase in tolerance was associated with a decrease in the lag time, which is the opposite of what has been observed in *E. coli* with ampicillin (and also opposite to what we would expect based on the rule of thumb that non-growing cells are better able to survive antibiotics).

Essential revisions:

The reviewers liked this work. They would like to see some clarifications and more clear writing - please see the points below.

*Reviewer #1:*

Aranda-Diaz et al., describe how microbial interactions alter the impact of antibiotics on bacteria found in the gut of fruit flies. The described mechanism of increasing tolerance by reducing lag time is novel.

One question I have is whether the increased tolerance could be the result of growth compensating for the number of dying cells or instead is purely a matter of cells dying at a slower rate. For example, if the cells were inoculated into carbon-free media and therefore could not replicate would the change in death rate still be observed?

I also think that there is room for clarifying the writing. For example, it would be helpful to have more information on how the bacterial strains were isolated. Were all strains isolated at the same time or were strains reported in previous papers isolated at different times. Did any of the strains come from the same fly? Similarly, it seems that asterisks are not used in a consistent manner in figures making it difficult to determine which results are significant.

Reviewer #2:

1) The figures in the manuscript are titled with Fig X or Figure X…), while the figure legends are titled with Figure X. Please change them to have consistent names (either Figure X or Fig.X) across the whole manuscript including supporting information.

2) Reformulate the first sentence of subsection “Co-culturing leads to growth of *Lp* in stationary phase”.

3) Please clarify the implication on human gastrointestinal tracts, which is more complex by studying fly gut microbiota in the Discussion section.

4) Discussion section, please be very careful with the definition of tolerance and resistance (as well as the differences between tolerance and resistance).

5) Discussion section, “or result from” should be changed to “or resulted from”.

6) Figure 1 legend, please indicate the meaning of the dashed line the exact MIC in Figure 1B or the figure legend and to make the figure self-understandable without reading the whole manuscript. Figure 1C legend, CFUs of *Lp* grown in monoculture (Lp) or in co-culture with Ax-cocultures was Normalized to CFUs at t=0? The figure legends in Figure 2C are confusing. Please revise it.

Reviewer #3:

The authors should do a more thorough search for published papers on pH and antibiotics to set the proper stage. An example is Karslake et al. (2016) (https://journals.plos.org/ploscompbiol/article?id=10.1371/journal.pcbi.1005098).

*Reviewer #4:*

I thought that this study illuminated a fascinating consequence of growth in the context of a multi-species community, and that many of the lessons could have broad implications. Moreover, I thought that the experiments were done with care and the conclusions were justified by the experiments and the analysis. If anything, I thought that two much space was devoted to exploring the pH dynamics and not enough to exploring the surprising finding of how/why a decrease in lag time would lead to an increase in tolerance.

We normally think of tolerance as being associated with an increase in the lag time, yet in this study the authors find the opposite. I would have appreciated some speculation in the Discussion section regarding why this might be, but I understand that the authors may not want to say anything that ends up not being true. However, in the Results section I think that the authors should be more clear about the direction of the effect and that this is a surprise (this is done in the Discussion section, but a short comment about it in the Results section may be appropriate). Also, if there were any experiments that the authors could do to illuminate this question then it would strengthen the paper.

I found the writing of the paper to be a bit dry and technical. Much of the Results section on pH interactions felt longer than it needed to be, with extended discussion of the precise values of experimental parameters and pH values. This is a stylistic question that should be left to the discretion of the authors, but the details at times obscured the central take-aways and made me lose some interest as I was reading.

Why are the percentage survival values in Figure 1C so much larger than in Fig1B? In Figure 1B at 20ug/mL it looks like only < 1 in 10^5 cells are surviving < 0.001%, whereas the survival rates in Figure 1C are 10%.

---

## [Author Response]

Essential revisions:Reviewer #1:[…] One question I have is whether the increased tolerance could be the result of growth compensating for the number of dying cells or instead is purely a matter of cells dying at a slower rate. For example, if the cells were inoculated into carbon-free media and therefore could not replicate would the change in death rate still be observed?

The reviewer brings up a very interesting point. To address this question, we imaged cells from an *Lp*+*Ap* co-cultureon MRS pads with rifampin. From these experiments, we did not observe any growth taking place, so it does not appear that the increased tolerance is due to a shift in the balance of growth and death. The new data is in Figure 1—figure supplement 2E.

We included the following text:

“To test whether the increased time to death was due to a disruption in the balance of growth and death, we performed single-cell imaging of mono- and co-cultures on MRS agarose with 50 µg/mL rifampin. We observed no growth in either mono- or co-cultures over 3 h of imaging (*n*=568 and *n*=236 cells, respectively; Figure 1—figure supplement 2D), indicating that survivability is due to protection from death, as opposed to increased growth. Because the MIC of *Lp* was unchanged in co-cultures compared with monocultures (no resistance), and the delay in killing was observed in the bulk population in co-cultures, these results indicate that co-culturing *Lp* with *Ap* induces tolerance of *Lp* to rifampin (Balaban et al., 2019; Brauner et al., 2016).”

I also think that there is room for clarifying the writing. For example, it would be helpful to have more information on how the bacterial strains were isolated. Were all strains isolated at the same time or were strains reported in previous papers isolated at different times. Did any of the strains come from the same fly?

We appreciate the reviewer’s insight into whether strains that are isolated together have more interactions. The strains used in this study were isolated from pools of flies and hence we cannot identify whether any came from the same animal. We note that these are the same strains that we have used in previous publications (Obadia et al.,2017 and Gould et al.,2018). This deep question about strain-level changes to interactions goes beyond the scope of this study, but we are actively investigating this idea.

Similarly, it seems that asterisks are not used in a consistent manner in figures making it difficult to determine which results are significant.

We apologize for the confusion. We have now corrected any inconsistencies and clarified any multiple hypothesis corrections.

Reviewer #2:1) The figures in the manuscript are titled with Fig X or Figure X…), while the figure legends are titled with Figure X. Please change them to have consistent names (either Figure X or Fig.X) across the whole manuscript including supporting information.

We have made these names consistent by using “Figure” throughout.

2) Reformulate the first sentence of subsection “Co-culturing leads to growth of Lp instationary phase”.

We have rewritten these lines. They now read:

“To investigate possible environmental factors that could be linked to *Lp* tolerance to rifampin when co-cultured with *Acetobacter* species (Figure 1C,E), we first inquired whether the total amount of growth of the co-culture was larger or smaller than expected from the yield of the monocultures.”

3) Please clarify the implication on human gastrointestinal tracts, which is more complex by studying fly gut microbiota in the Discussion section.

We have clarified the implications of our results on the human GI tract by adding the following text:

“Our findings linking short chain fatty acid metabolism, growth, and antibiotic action in commensal microbes from the fruit fly gut opens the door to studying these phenomena in a model organism. While the human gut microbiome comprises hundreds of bacterial species, the simplicity of the *Drosophila* gut microbiota (Wong et al., 2011), the genetic tractability of *Drosophila*, and the fact that ~65% of human disease-causing genes have homologs in the *Drosophila* genome (Ugur et al., 2016), make the fruit fly a powerful model for host-microbiome interactions (Douglas, 2018).”

4) Discussion section, please be very careful with the definition of tolerance and resistance (as well as the differences between tolerance and resistance).

We appreciate the reviewer’s point and have been very careful about the definitions in our manuscript by reviewing a recent review by Balaban et al., that seeks to clarify definitions of persistence and tolerance. We have modified the following paragraph in the Introduction:

“Bacteria can survive antibiotics through (i) resistance mutations, which counteract the antibiotic mechanism and increase the minimum inhibitory concentration (MIC); (ii) tolerance, whereby the entire population enters an altered physiological state that prolongs survivability without changing the MIC of the antibiotic, leading to an increase in the time required to kill a given fraction of the population; (iii) heteroresistance, whereby a subset of the population has a higher MIC and grows at concentrations that would otherwise kill the population; and (iv) persistence, whereby a subset of the population survives treatment for a longer period (Balaban et al., 2019; Brauner et al., 2016).”

The lines mentioned by the reviewer now read:

“In this study, we observed a novel form of antibiotic tolerance. Tolerance has been defined as increased time to killing of the population as a whole (Brauner et al., 2017), as opposed to resistance (a change in the MIC), or persistence or heterotolerance whereby a subpopulation of bacteria displays increased time to killing (Balaban et al., 2019; Brauner et al., 2016).”

5) Discussion section, “or result from” should be changed to “or resulted from”.

We have made this change.

6) Figure 1 legend, please indicate the meaning of the dashed line the exact MIC in Figure 1B or the figure legend and to make the figure self-understandable without reading the whole manuscript.

We apologize for the confusion, we have added a label to the dashed line to the figure and clarified the legend.

Figure 1C legend, CFUs of Lp grown in monoculture (Lp) or in co-culture with Ax-cocultures was Normalized to CFUs at t=0?

The reviewer is correct, and we have explained the normalizations in all legends more clearly.

The figure legends in Figure 2C are confusing. Please revise it.

We have revised this legend.

Reviewer #3:I have already provided my critiques during the pre-submission stage. The authors should do a more thorough search for published papers on pH and antibiotics to set the proper stage. An example is Karslake et al. (2016) (https://journals.plos.org/ploscompbiol/article?id=10.1371/journal.pcbi.1005098).

We appreciate all of the reviewer’s feedback. We have included the reference above, and others to set the stage for our discussion of pH and antibiotics. The following paragraph was modified in the Introduction:

“For example, the intimate relationship between bacterial metabolism and environmental pH could also lead to changes in antibiotic efficacy, as previously shown in monocultures (Aagaard et al., 1991; Argemi et al., 2013; Kamberi et al., 1999; Karslake et al., 2016; Yang et al., 2014). The interplay of all of these processes in complex communities will provide new ways to combat pathogen survival and resistance evolution, particularly in cases involving tolerance, an important and understudied aspect of antibiotic susceptibility that can be elicited by diverse mechanisms and can facilitate the evolution of resistance (Levin-Reisman et al., 2017).”

Reviewer #4:I thought that this study illuminated a fascinating consequence of growth in the context of a multi-species community, and that many of the lessons could have broad implications. Moreover, I thought that the experiments were done with care and the conclusions were justified by the experiments and the analysis. If anything, I thought that two much space was devoted to exploring the pH dynamics and not enough to exploring the surprising finding of how/why a decrease in lag time would lead to an increase in tolerance.We normally think of tolerance as being associated with an increase in the lag time, yet in this study the authors find the opposite. I would have appreciated some speculation in the Discussion section regarding why this might be, but I understand that the authors may not want to say anything that ends up not being true.

We appreciate the reviewer’s comments on our study, and we have added some speculation about the link between lag time and tolerance, while emphasizing that our work highlights the inherent complexity of the tolerance phenotype.

We have added the following text to the Discussion section:

“The opposite connections with rifampin and erythromycin tolerance underscore the complexity of the link between growth and antibiotic action. Changes in growth may lead to changes in the levels and activity of these antibiotics’ molecular targets. Moreover, the molecular mechanisms that lead to death downstream of the antibiotic target could be a function of the growth state of the cell.Previous work has shown that bacterial interactions can elicit changes in antibiotic sensitivity by changing cellular physiology or interfering with antibiotic action directly or indirectly (Adamowicz et al., 2018; Radlinski et al., 2017; Sorg et al., 2016).”

However, in the Results section I think that the authors should be more clear about the direction of the effect and that this is a surprise (this is done in the Discussion section, but a short comment about it in the Results section may be appropriate).

We have added this point to the Results section in the following paragraph:

“We previously observed for *Lp* monocultures when shifting the pH that the stimulation of growth in stationary phase was connected with a shorter lag phase (Figure 3E).In agreement with these data, there was a significant decrease in bulk lag time for all of the *Acetobacter* co-cultures (Figure 4A,B). Surprisingly, this *shorter* lag phase was linked to *Lp* tolerance to rifampin, opposite to the longer phase linked to *E. coli*tolerance to ampicillin (Brauner et al., 2016).”

Also, if there were any experiments that the authors could do to illuminate this question then it would strengthen the paper.I found the writing of the paper to be a bit dry and technical. Much of the Results section on pH interactions felt longer than it needed to be, with extended discussion of the precise values of experimental parameters and pH values. This is a stylistic question that should be left to the discretion of the authors, but the details at times obscured the central take-aways and made me lose some interest as I was reading.

We appreciate the reviewer’s interest, which we share. We were not able to come up with any experiments to illuminate this question, but we agree that it is a very exciting future research direction.

Why are the percentage survival values in Figure 1C so much larger than in Figure 1B? In Figure 1B at 20ug/mL it looks like only < 1 in 10^5 cells are surviving < 0.001%, whereas the survival rates in Figure 1C are 10%.

We apologize for the confusion. The numbers in Figure 1B are measured relative to the initial inoculum, and for 20 µg/mL the total number of viable counts (CFUs) is ~0.1 or ~0.01 of the initial number of cells for *Lp*+*Ap* and *Lp* alone, respectively(Figure 1B). These numbers are in good agreement with the data in Figure 1C, which shows that 10% and 1% survival for *Lp*+*Ap* and *Lp*, respectively.